



# A storyline-based approach towards changing typhoon intensities over the Pearl River Delta under future conditions using Pseudo-Global Warming

Patrick Olschewski[1]*, Qi Sun[1], Jianhui Wei[1], Yu Li[1], Zhan Tian[2,3], Laixiang Sun[4,5], Joël Arnault[1], Tanja C. Schober[1], Brian Böker[1,6], Harald Kunstmann[1,7,8], Patrick Laux[1,7]

[1]Institute of Meteorology and Climate Research – Atmospheric Environmental Research (IMKIFU), Karlsruhe Institute of Technology, Garmisch-Partenkirchen, 82467, Germany
[2]School of Environmental Science and Engineering, Southern University of Science and Technology, Shenzhen, 518055, People's Republic of China
[3]Pengchen Laboratory, Shenzhen, 518066, People's Republic of China
[4]Department of Geographical Sciences, University of Maryland, College Park, MD 20740, United States of America
[5]School of Finance & Management, SOAS University of London, London, WC1H 0XG, United Kingdom
[6]Dr. Blasy – Dr. Øverland Ingenieure GmbH, Eching am Ammersee, 82279, Germany
[7]Institute of Geography, University of Augsburg, Augsburg, 86159, Germany
[8]Centre for Climate Resilience, University of Augsburg, Augsburg, 86159, Germany

*Correspondence to*: Patrick Olschewski (patrick.olschewski@kit.edu)

**Abstract.** During the Pacific typhoon season of 2023, the Pearl River Delta (PRD) was hit by a series of typhoons, causing unprecedented precipitation as well as fatalities and significant damages. There are indications that these events may intensify under climate change. However, the unfolding of similar events in the future is yet to be fully understood. We therefore conducted an investigation of historical typhoon events affecting the PRD using Pseudo-Global Warming. Within this framework, we perturbed the historical initial and boundary conditions obtained from ERA5 reanalysis and handed to the regional model WRF according to the climate change signal projected by the CMIP6 ensemble under SSP5-8.5. We pursued a storyline-based approach, in which each of the 16 selected CMIP6 models acted as the basis for an individual storyline of future typhoon intensity. Additionally, we created two storylines based on thermodynamic drivers to create scenarios with excessively favorable and unfavorable conditions for typhoon intensification, resulting in 18 storylines that were assessed for seven representative typhoons. WRF was set up in a two-way nesting framework using domains of 25 km and 5 km resolution, of which the latter was used for further assessment. For each typhoon, the simulations were initiated at the start of the first intensification phase and statistically assessed until landfall. Minimum sea level pressure, maximum wind speed, mean and maximum 1-hourly precipitation rates as well as the integrated kinetic energy (IKE) as an advanced measure for typhoon damage potential were used to determine typhoon intensity. Results indicate a general increase in typhoon intensity across all metrics for six of the seven inspected typhoons. This increase is notably higher for specific storylines, and the projected increase in the extreme values of the inspected metrics significantly exceed the median change of all storylines. This indicates that the true potential range may lie above what would be expected under a median approach. The results suggest a maximum decrease of up to 15 hPa for minimum central pressure, 11 m s$^{-1}$ increase for maximum wind speed, 2.5 mm h$^{-1}$ for mean and





mm h$^{-1}$ for maximum precipitation rates. The two additional storylines revealed an even higher intensity increase in the form of central minimum pressure decreases and wind maxima increases for two typhoons, but mostly resembled the span provided by the 16 GCM-based storylines. These results can support the optimization of the development of protective measures considering the improved range of intensity potential.

## 1 Introduction

Tropical cyclones are one of the biggest natural risk factors for coastal areas in the tropics and subtropics across the globe. Connected with high wind speeds, torrential rain, and flooding, it is especially the compound nature of these extremes that make tropical cyclones a substantial socioeconomical threat. On average, 20.4 million people worldwide are affected by tropical cyclones with losses reaching into the billions of US$ (EM-DAT The International Disaster Database., 2024). In its latest Assessment Report, the International Panel on Climate Change (IPCC) emphasized that the mechanisms of future tropical cyclone development remain to be fully understood (Seneviratne et al., 2021). Nonetheless, the IPCC was able to attribute high confidence to tropical cyclones imposing a detrimental factor to economic growth and increasing levels of precipitation due to human-caused climate change (Calvin et al., 2023). In addition, due to their destructive potential, tropical cyclones are listed as one of the main reasons for involuntary migration, and in global comparison, residents of South, Southeast and East Asia are most affected (Pörtner et al., 2022). In the wake of this, however, there is only medium confidence in findings of increased intensities of tropical cyclones under global warming (Calvin et al., 2023), pointing out the immediate necessity for further investigations.

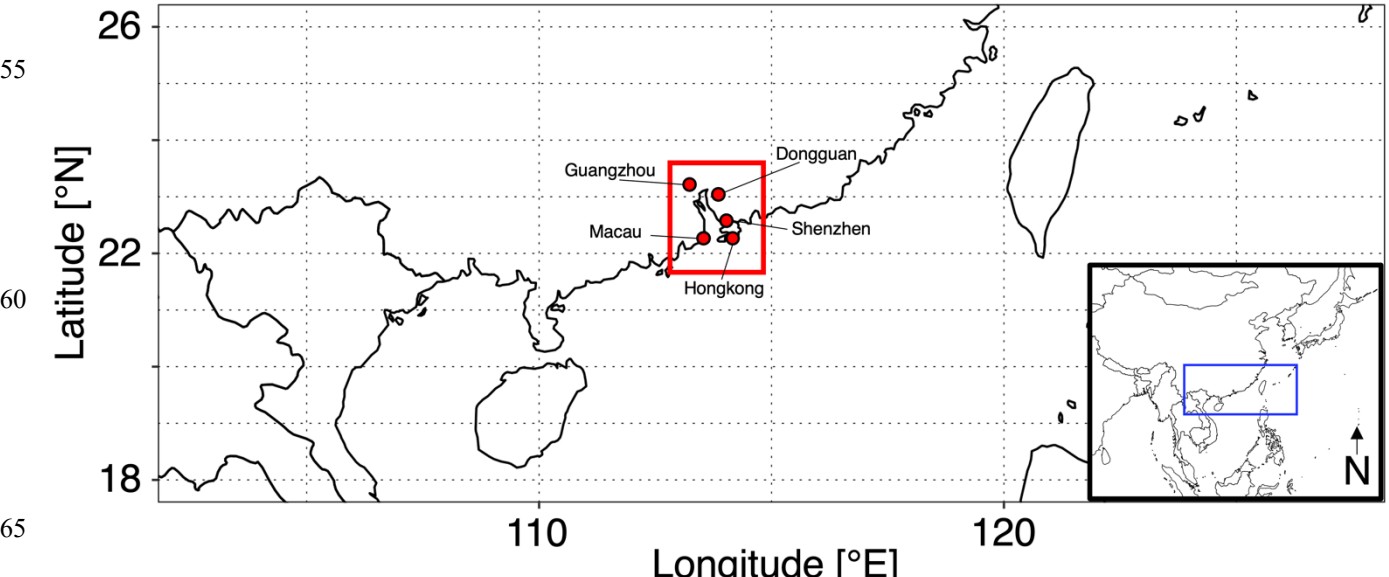

Figure 1: Study domain, the Pearl River Delta (PRD).



In highly vulnerable regions an improved understanding of the potential unfolding of tropical cyclones under climate change
is all the more important. Next to the aforementioned risk of destruction-cased displacement, maintaining critical infrastructure, as well as the prevention of deaths and economic losses are aggravated in regions with high population density and economical and political importance. One of these regions is the Pearl River Delta (PRD), which is located in Southeast China, within the Guangdong province. The PRD is home to established metropolises and rapidly growing megacities such as Guangzhou, Shenzhen, and Dongguan, as well as the special administrative regions Hong Kong and Macau (Fig. 1). According to Wan et
al. (2023), Guangdong was the most frequently affected Chinese province in terms of severe tropical cyclones, in this region referred to as typhoons, between 2009 and 2019. Between August and October 2023, the PRD was affected by a series of typhoons, causing unprecedented rainfall in Hong Kong, as well as dozens of injuries and substantial damages across the PRD, reaching far into the millions of US$ (Kwan, 2023). There are indications within former studies that typhoons affecting Southeast China may intensify due to global warming (Huang et al., 2022; Chen et al., 2020, 2021), which could make the
events of 2023 a precedent for the unfolding consequences of climate change. The present study is therefore conducted to improve our knowledge on landfalling typhoons affecting the PRD in a warming climate.

With regards to the question how extreme weather and climate events may change in warming climate, Shepherd (2016) introduced storylines as an alternative to a probabilistic risk approach. It was shown that multiple factors determining the outcome of an investigated event, such as thermodynamical and dynamical aspects of the atmospheric state, can be individually
assessed when each potential state of these factors is assigned a storyline. Shepherd et al. (2018) defined the plausibility of changes in these factors to be the main focus of assessment within a storyline-based framework, rather than their probability. Within this framework, as Shepherd (2016) pointed out, a historical typhoon event can serve as a benchmark and its unfolding in a warmer climate may be assessed while also accounting for uncertainties in the magnitude of change in each of the thermodynamical and dynamical drivers.

One way of assessing changes in extreme weather and climate events within a storyline-based framework is the Pseudo-Global Warming approach (PGW, Shepherd (2019), Hazeleger et al. (2015)). Originally presented by Schär et al. (1996), PGW is a computationally cost-friendly alternative to conventional long-term modelling. Instead of running a model for a long period of time, e.g., 30 years, and assessing the results on a probabilistic basis, meteorological hazards are simulated on an event-to-event basis, while the initial and boundary conditions of the model are perturbed according to a climate change signal. For
example, reanalysis data for specific typhoons can serve as boundary and initial conditions to run a regional climate model for these typhoons under the corresponding historical climate conditions. A climate change signal is then added to the reanalysis data and the model is re-run with these perturbed initial and boundary conditions, enabling a comparison of the simulations under historical and future conditions.

Several studies focusing on historical typhoon events affecting the PRD under potential future climate conditions were recently
published, for example by (Chen et al., 2020, 2022; Chow et al., 2024). These studies have the use of an ensemble mean of CMIP5/CMIP6 models to perturb the initial and boundary conditions of the model in common. While it is a reasonable





approach to use an ensemble mean in order to reduce the sensitivity to large deviations among the ensemble members (Rasmussen et al., 2011; Redmond et al., 2015), the resulting range of the possible outcome is significantly limited when only one scenario of future climate is investigated (Räisänen and Palmer, 2001). For the PRD, to the best of our knowledge, there

is a lack in studies extensively studying future intensities of typhoons using the climatology of an ensemble of GCMs as input for PGW. In addition, the use of the latest CMIP6 ensemble instead of the former CMIP5 ensemble represents an advancement compared to former studies. In a recent study for the Philippines, Delfino et al. (2023) explored future typhoon intensities using a PGW framework while accounting for multiple models within the CMIP6 model ensemble. In extending this approach to the PRD and using an increased number of CMIP6 models, we aim to offer deeper insights into the variability of typhoon

intensities over the PRD under a warming climate. In doing so, we define each of the included models of the CMIP6 ensemble as an individual storyline that is equally treated in the subsequent assessment of typhoon uncertainty.

As Lin and Emanuel (2016) pointed out, even when considering multiple storylines of how the climatic conditions may evolve, cases of an unprecedented unfolding of events that was historically considered to exceed the given limits may go undetected. Within former studies, each PGW experiment was conducted using the climatology of a specific model or the ensemble mean

(Chow et al., 2024; Delfino et al., 2023; Chen et al., 2020), limiting the results to the physical limits imposed by the chosen model for PGW. However, as Shepherd (2016) pointed out, an advantage of the storyline-based approach is the ability to assess each involved driver individually. Within our study, we opt to use this advantage to further explore the physical limits of typhoon sensitivity. We reassemble the individual temperature-based drivers of the CMIP6 models to create two additional storylines with a low increase in thermodynamic stability and a high level of sea surface warming and vice-versa to resemble

excessively favorable and excessively unfavorable conditions for typhoon intensification. This is based on the findings of Shen et al. (2000), Hill and Lackmann (2011), and Tuleya et al. (2016) who found that an increase in thermodynamic atmospheric stability and increased sea surface temperatures counteract with regards to typhoon intensity,

With the help of the above-presented framework, we seek to answer the following research questions: a) How well are historical typhoons affecting the PRD represented by our selected model set up regarding track accuracy and intensity? b) To

what extent are these typhoons projected to change in intensity under 16 storylines of a high-impact climate scenario (SSP5-8.5)? c) To what extent do typhoon intensities exceed the range provided by the 16 storylines, when thermodynamical drivers of these storylines are reassembled to create excessively favorable and unfavorable conditions for typhoon intensification?

## 2 Data and Methods

### 2.1 Data

The regional climate model was driven by ERA5 reanalysis data obtained from the European Centre for Medium-Range Weather Forecasts (ECMWF, Hersbach et al. (2020)). This latest version of reanalysis products by the ECMWF provides over 200 variables for atmospheric pressure levels (Hersbach et al., 2023a) and the surface level (Hersbach et al., 2023b), from 1940 until the present, and on an hourly basis for a spatial resolution of 31 km, resp. 0.25° x 0.25°. The corresponding month



of data was obtained for each selected case and used as initial and boundary conditions for the regional climate model, in the

case of the PGW after the corresponding modification.

The climate change signal that is applied within PGW was obtained from the CMIP6 model archive, an initiative by the World Climate Research Programme (WCRP) that is currently in its sixth generation (Eyring et al., 2016) . Based on the availability of all the necessary variables, an ensemble of 16 GCM models is selected for this study and is in detail described in Table 1.

**Table 1. 16 models obtained from CMIP6 for the derivation of PGW signals.**

| Model name | Institution | Source |
|---|---|---|
| AWI-CM-1-1-MR | Alfred Wegener Institute | Semmler et al. (2020) |
| BCC-CSM2-MR | Beijing Climate Center | Wu et al. (2019) |
| CAMS-CSM1-0 | Chinese Academy of Meteorological Sciences | Rong et al. (2018) |
| CanESM5 | Canadian Centre for Climate Modelling and Analysis | Swart et al. (2019) |
| CMCC-CM2-SR5 | Euro-Mediterranean Centre on Climate Change | Cherchi et al. (2019) |
| CMCC-ESM2 | Euro-Mediterranean Centre on Climate Change | Lovato et al. (2022) |
| EC-Earth3-Veg-LR | European Research Consortium | Döscher et al. (2022) |
| INM-CM4-8 | Russian Academy of Science | Volodin et al. (2018) |
| INM-CM5-0 | Russian Academy of Science | Volodin et al. (2017) |
| KIOST-ESM | Korea Institute of Ocean Science and Technology | Pak et al. (2021) |
| MPI-ESM1-2-HR | Max Planck Institute for Meteorology | Gutjahr et al. (2019) |
| MPI-ESM1-2-LR | Max Planck Institute for Meteorology | Fiedler et al. (2019) |
| NESM3 | Nanjing University of Information Science and Technology | Cao et al. (2018) |
| NorESM2-LM | NorESM Climate Modeling Consortium | Seland et al. (2020) |
| NorESM2-MM | NorESM Climate Modeling Consortium | Seland et al. (2020) |
| TaiESM1 | Research Center for Environmental Changes, Taiwan | Lee et al. (2020) |

Monthly data from each model was obtained for the SSP5-8.5 scenario and the 18 vertical layers (17 pressure levels + near surface layer) that are available across all 16 models. Subsequently, the mean difference between the future period 2071-2100 and the historical period 1985-2014 was calculated and used as climate change delta within the PGW process. Correspondingly, 18 vertical layers were obtained from ERA5, modified in accordance with the CMIP6-derived climate change delta, and subsequently handed to the regional climate model.

For the performance assessment of the historical simulations, two reference data sets of historically tracked typhoon information were obtained. The CMA Tropical Cyclone Best Track Dataset is provided by the China Meteorological Administration (Lu et al., 2021; Ying et al., 2014) and offers 6-hourly information on the location, minimum central pressure, and maximum wind speeds of Northwest-Pacific typhoons from 1949 until the present. The Best Track Data by the Japan Meteorological Agency (JMA), i.e. the Regional Specialized Meteorological Center (RSMC) Tokyo - Typhoon Center

(RSMC), offers the same information, next to additional details on radii of specific wind speeds, from 1951 until the present (RSMC Tokyo - Typhoon Center, 2023). Both datasets were obtained and used in this study, in order to additionally highlight uncertainties arising from the choice of reference. Out of the historical track data, 28 typhoon events that affected the Pearl River Delta were selected for the initial performance assessment of the regional climate model. Based on the performance of WRF in representing the historical intensity of central minimum pressure and maximum wind speed and to cover different

intensity categories based on the Saffir-Simpson Hurricane Wind Scale (SSHWS, Saffir 1973), seven events were selected and subsequently processed within the PGW framework of this study. We then added precipitation levels and an advanced measure of typhoon damage potential, i.e., integrated kinetic energy (IKE), for the assessment of the PGW-based simulations.



Since precipitation measurements over the ocean are scarce and additional limitations would have been imposed when using radar or satellite measurements, we limited the validation of our historical simulations to track accuracy, central minimum
pressure and maximum wind speed. This data is consistently provided within the two selected reference data sets and this form of validation is frequently accepted within tropical cyclone research, yet limiting the interpretability of the precipitation results.

**2.2 Climate Change Signal**

The climate deltas derived from the 16 CMIP6 GCM models and used for the perturbation of the initial and boundary
conditions from ERA5 are applied on a grid-cell basis. According to the findings of Xue et al. (2023) this is the preferable option compared to using a spatial mean. In order to simplify the preliminary assessment of the 16 selected GCMs, however, we inspected the spatial mean values within a standardized area. This area within 10° N to 20° N and 110° E to 170° E is essential for cyclogenesis and hence defined by Chiacchio et al. (2017) as the West Pacific tropical cyclone development region (WP-TCDR). Among all atmospheric variables that play a role in tropical cyclogenesis, temperatures are crucial. It was
found that sea surface temperatures highly correlate with the destructive potential of tropical cyclones (Emanuel, 2005) and that increased sea surface temperatures due to global warming could exacerbate hurricane intensity (Duan et al., 2018). It was also found that an increased thermodynamic stability in the atmosphere under the current climate projections counteracts a potential intensity increase due to elevated levels of sea surface temperature (Shen et al., 2000; Hill and Lackmann, 2011; Tuleya et al., 2016). We therefore assess the climate change signals of the 16 GCMs based on the sea surface and atmospheric
temperatures and the results are displayed in Fig. 2.

The projected spatial average in temperature increase for the 17 vertical pressure levels over the WP-TCDR are shown in Fig. 2a. Only the months of the seven selected typhoon events are shown. Across all models, the magnitude of atmospheric warming increases upward from around 2° C – 5° C on the 1000 hPa level to around 4° C – 10° C on the 200 hPa level. Above the tropopause, the level of warming decreases and a cooling is projected for the stratosphere above the 70 hPa level. All models
therefore project an increase in thermodynamic stability within the tropopause. However, the magnitude of this stabilization varies greatly. Models projecting a high sea surface temperature increase also project a high gradient in the level of warming near the surface and near the tropopause, and vice versa.

Figure 2b depicts the spatial average in sea surface temperature increase over the WP-TCDR. Depending on the selected GCM, the level of warming reaches from 1.1° C to 4.4° C. In contrast to atmospheric temperatures, the inter-monthly variations are
high. When inspecting all models, a stronger increase towards earlier months within the year may be assumed, but only few models actually show this tendency, i.e., CanESM5, INM, and KIOST-ESM. Distinct, on the other hand, is the link of a high level of atmospheric warming to a high level of sea surface temperature warming. Based on this link and in comparison to the findings of Emanuel (2005), our selection of the 16 GCMs with all necessary variables available obviously includes those GCMs of the CMIP6 ensemble that show both the highest and lowest projected change signals for sea surface temperature for





the temperature-based PGW assessment. This confirms that our choice is a suitable subset of CMIP6 GCMs for the analysis

of future, storyline-based typhoon intensities under PGW.

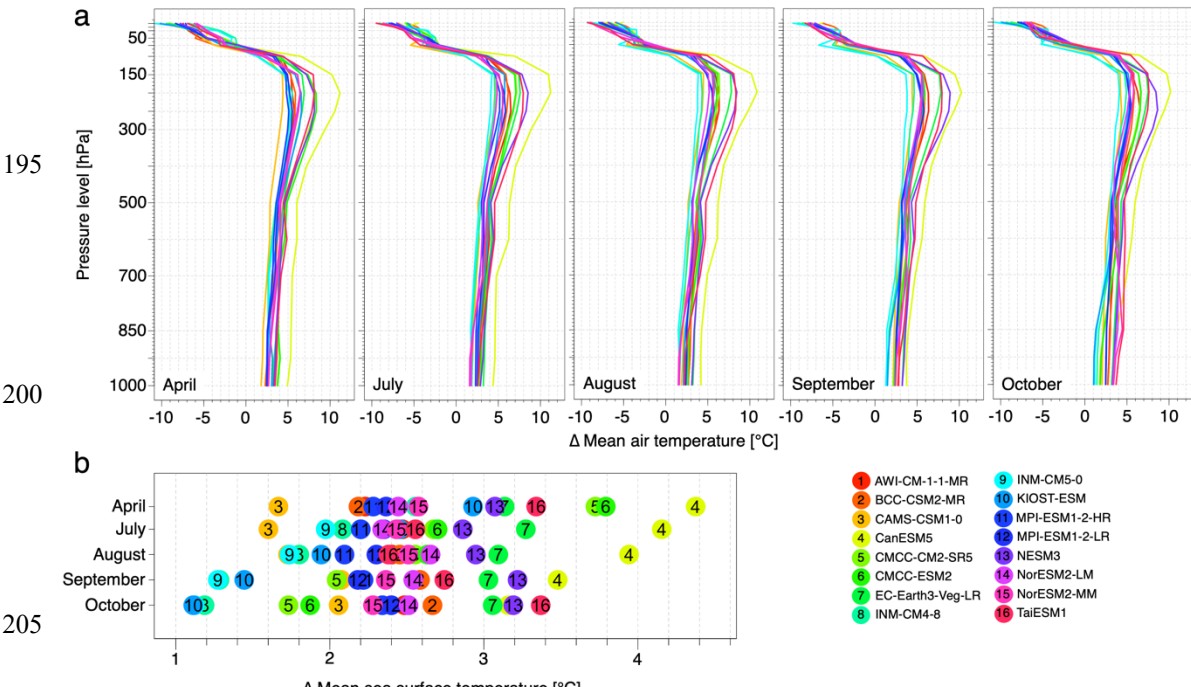

**Figure 2: Projected spatial mean temperature changes within the West Pacific tropical cyclone development region (WP-TCDR). a) projections of atmospheric temperature changes for the 17 vertical levels provided by the 16 GCMs, b) projections of sea surface temperature increases. Only the months of the seven selected typhoon events are shown.**

**2.3 Methods**

The simulations are performed using version 4.3.3 of WRF-ARW, the Advanced Research Weather Research and Forecasting

Model (Skamarock et al., 2019). While there exist models with increased capabilities in simulating hurricane intensity, for

example HWRF (Bernardet et al., 2015; Goldenberg et al., 2015), we are mainly interested in the general unfolding of typhoons

rather than highly precise nowcasting. We therefore proceed with WRF-ARW which is proven to perform sufficiently within

similar research frameworks and is less cost-intensive (Gutmann et al., 2018; Chen et al., 2020; Delfino et al., 2023). While

many studies demonstrated the abilities of WRF in the field of typhoon research for the Northwest Pacific (Delfino et al., 2022;

Shirai et al., 2022; Lui et al., 2021; Cha et al., 2011), the selection of model parameterization schemes remains complex and

depends on the research intentions, the selected level of acceptable uncertainties and the selected events (Sun et al., 2024). In

addition, the application of PGW imposes additional uncertainties on the modeling framework. Therefore, we adapted an

acknowledged WRF framework used by Delfino et al. (2023) and Olschewski and Kunstmann (2024) in applications for the

eastern United States and an adjacent region to this study in the Northwest Pacific. Sun et al. (2024) conducted an extensive

investigation of optimized initialization times which was adopted in this study. Therefore, the simulations were initialized

when the typhoon first reached Tropical Storm intensity (maximum wind speed > 17 m s$^{-1}$). While the inspection of a single





model realization may limit the results of the study, the main focus of this study was intentionally shifted to the uncertainties

among the 18 storylines, using a parameterization set up (Delfino et al., 2023) and an initialization set up (Sun et al., 2024) that previously proved to perform well for similar research frameworks within typhoon modeling. Additionally, we expect the relative differences of these results to behave orthogonally, i.e. the variations in typhoon intensity to behave in a similar manner for different model set ups, following Xue et al. (2023). Within the WRF set up, two domains are simulated in a two-way nesting framework using a 25 km and 5 km resolution, as displayed in Fig. 3 with grids of 295 x 160 grid points and 926 x 551

grid points, respectively. We consider 52 vertical eta levels and the 50 hPa level was set as atmospheric top. The WRF Single-Moment 6-class scheme (WSM6) is used as microphysics scheme (Hong and Lim, 2006), the Kain-Fritsch scheme for cumulus parameterization (Kain, 2004), and the Yonsei University scheme (YSU) as planetary boundary layer scheme (Hong et al., 2006). The Rapid Radiative Transfer Model (RRTM) is used for longwave radiation (Mlawer et al., 1997) and the Dudhia scheme for shortwave radiation (Dudhia, 1989), as well as the Noah Land Surface Model as surface scheme (Chen and Dudhia,

2001) and the MM5 Monin–Obukov scheme as surface layer (Monin and Obukhov, 1954).

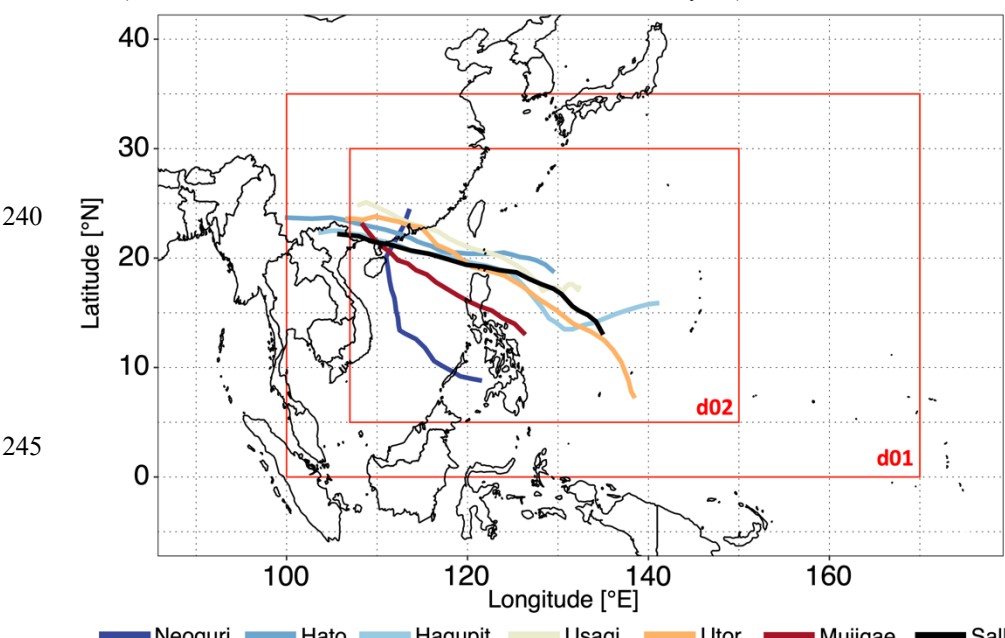

**Figure 3: Study region and outlines of two domains simulated in WRF. Colored paths indicate historical tracks of the seven selected typhoon events.**

In addition, a spectral nudging scheme is applied to align the simulated typhoon tracks to the historical tracks. On a 6-hourly basis, horizontal wind speeds above the 500 hPa level are spectrally nudged, i.e., a model-interior forcing aligning the model calculations closer to the ERA5 input data is applied (von Storch et al., 2000). This results in a sufficient alignment of the

tracks, but does not influence the inner core of the simulated typhoon.

In this study, two different configurations of PGW are investigated. On the one hand, a comprehensive perturbation of the initial and boundary conditions is performed for the 16 GCM-based simulations of each typhoon event. Within this set up,





atmospheric temperature, relative humidity, geopotential and horizontal winds are perturbed for each pressure level, as well as skin temperature, sea surface temperature, surface pressure and sea level pressure. However, as Xue et al. (2023) and

Olschewski and Kunstmann (2024) demonstrated, the results of PGW experiments may be highly sensitive towards which variables are perturbed within the PGW configurations. Therefore, we additionally assess the sensitivity of the selected cases towards the purely temperature-based change signals (atmospheric temperature, skin temperature, sea surface temperature) projected by the 16 GCMs. Based on the findings of Shen et al. (2000), who state that atmospheric warming and surface warming act as antagonists with regard to typhoon intensities, we select the largest single-variable change signal for sea surface

temperature out of the 16 selected GCMs and combine it with the lowest single-variable change signal for atmospheric temperature. Vice versa, the second sensitivity experiment is driven by the lowest single-variable change signal for sea surface temperature and the largest single-variable change signal for atmospheric temperature. Accordingly, within these two sensitivity experiments, only temperature is modified to create experimental situations in which a high sea surface temperature anomaly is aligned with a low level of atmospheric warming and vice versa, thus creating thermodynamically-based storylines

with excessively favorable and unfavorable conditions for typhoon intensification.

To deepen the understanding of the projected range of intensity, we apply an advanced measure for the typhoon damage potential. Based on (Morris and Ruf, 2017), we calculate the integrated kinetic energy (IKE) of a typhoon at a given timestep as a function of the storm size and the wind speed. IKE is therefore defined as the volumetrically integrated surface wind speed $U$ for a given air density $\rho$, for all grid points exceeding the defined wind speed threshold of 18 m s[-1](Kreussler et al., 2021).

For our study, we simplify $\rho = 1$ kg m[-3] (Kreussler et al., 2021) calculate the IKE as given in Eq. (1):

$$IKE = \int_V \frac{1}{2} \rho U^2 \, dV \quad (1)$$

Since there exist indications that the traditional Saffir-Simpson Hurricane Wind Scale may be insufficient to determine the potential impact of tropical cyclones (Kantha, 2006; Bloemendaal et al., 2021; Wehner and Kossin, 2024), we selected the IKE to better capture the true damage potential of the inspected typhoons.

To perform the statistical assessment of the simulated typhoon events, a tracking algorithm presented by Gutmann et al. (2018) is adopted. When the sea level pressure of a grid cell is 27 hPa or more below the 30-year maximum value and the corresponding maximum wind speed within a 400 km x 400 km box around this grid cell is higher than 25 m s[-1], tracking is initiated. If the local minimum within this box remains at least 17 hPa below the long-term maximum and the maximum wind speed above 15 m s[-1], tracking is continued and the new local minimum defined as typhoon center for the corresponding time

step. All statistics retrieved within the tracking algorithm, i.e., minimum central pressure, maximum wind speed, 1-hourly mean and maximum precipitation rate, refer to the 400 km x 400 km box around the storm center.





## 3 Results

### 3.1 Historical performance evaluation of simulated typhoons

Based on minimum central pressure, maximum wind speed, and the track accuracy, the 28 typhoon cases are evaluated with
regard to the mean temporal bias. Figure 4 depicts the three investigated metrics with respect to the two reference data sets. In general, a linear relationship between the bias of minimum central pressure and that of wind speed can be detected. However, the spread is higher for the JMA reference data set. Maximum wind speed is overestimated for the majority of events, but only rarely by more than 10 m s$^{-1}$ (e.g., no. 6, York), whereas the events spread equally within a range of -10 to +10 hPa for minimum central pressure. Exceptions with regard to minimum central pressure are the most intense events, i.e., Haima (no.
20) and Mangkhut (no. 24), which are significantly underestimated by the model. Storms with a higher intensity, resp. a higher typhoon category, in general show lower deviations of the simulated and historical tracks. By implication, the highest deviations are detected for the Tropical Storms and Category 1 typhoons, but do not exceed 100 km.

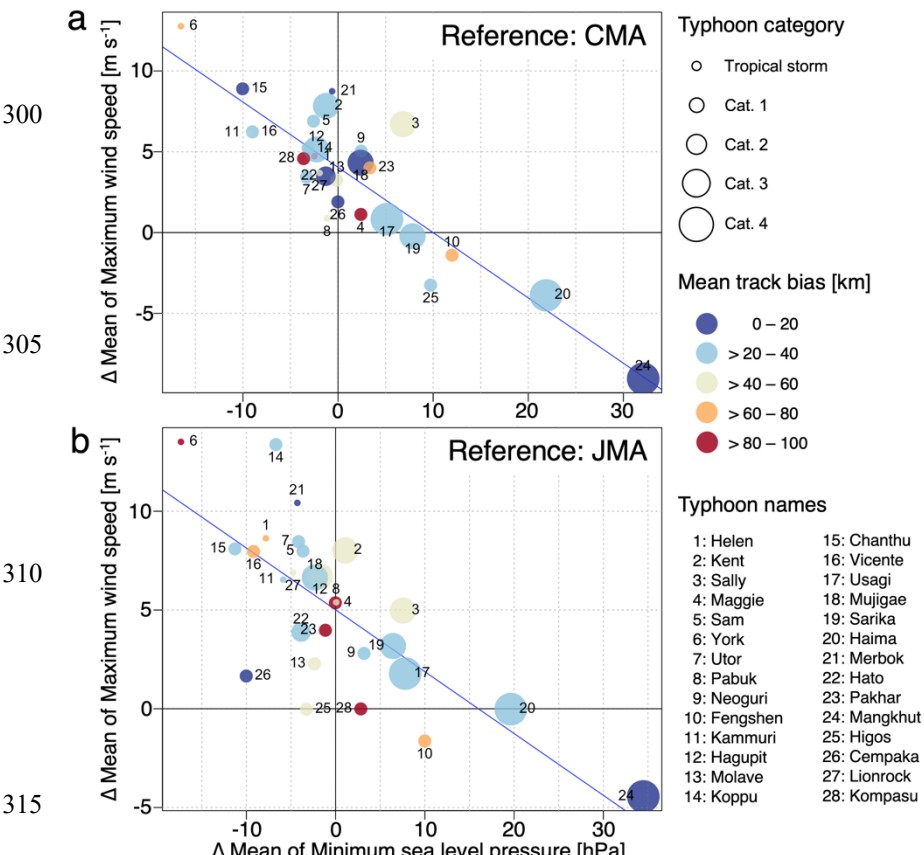

**Figure 4: Mean bias of minimum central pressure and maximum wind speed for 28 typhoon events affecting the Pearl River Delta, calculated as WRF minus reference. Historical storm intensity based on the SSHWS represented by circle radii and mean track bias displayed by the color scheme. a) all deviations with reference to the CMA reference data, b) all deviations with reference to the JMA reference data.**

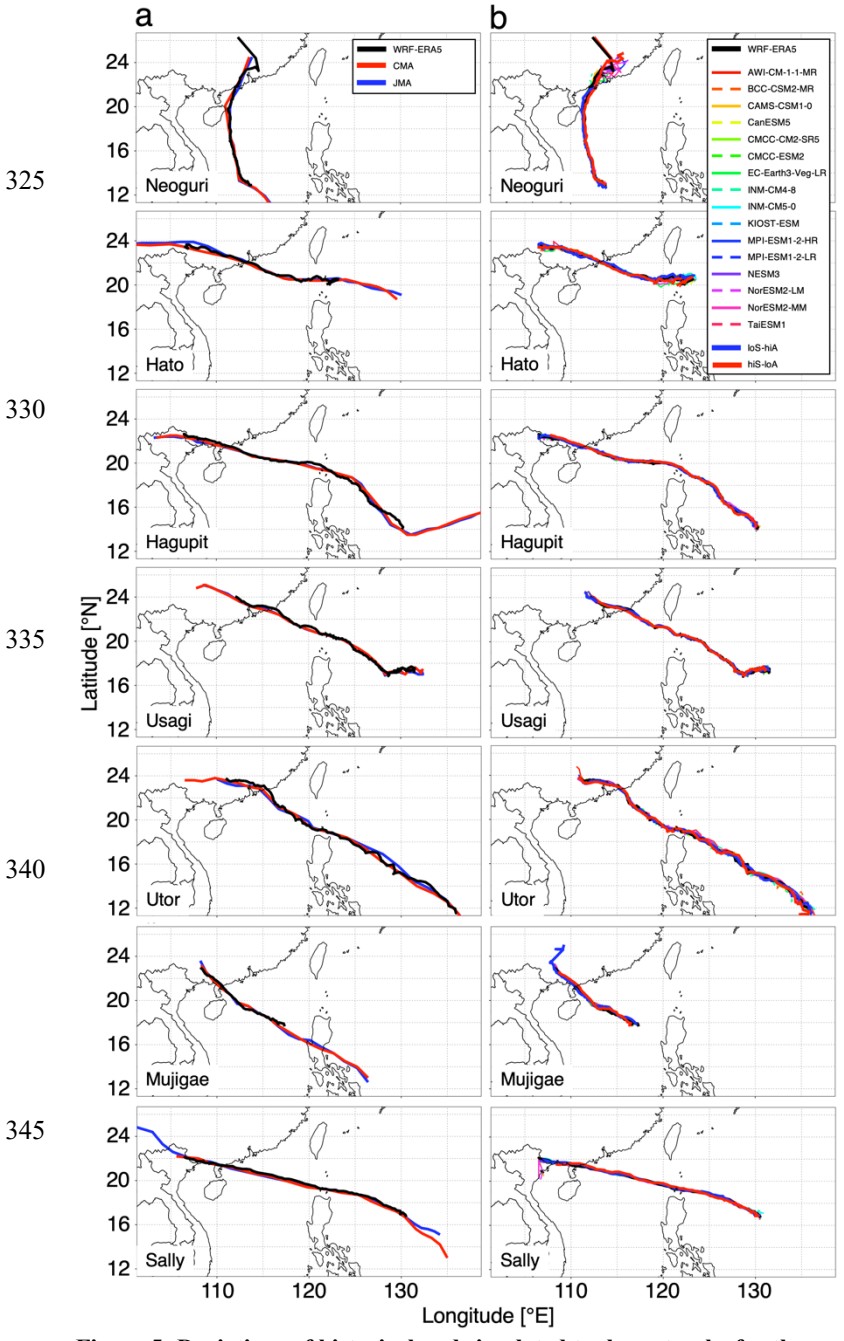

**Figure 5: Deviations of historical and simulated typhoon tracks for the seven investigated events. a) historical tracks derived from the CMA and JMA references and simulated tracks of the reanalysis-driven runs, b) simulated tracks of the reanalysis-driven runs, the 16 selected GCMs for the comprehensive PGW modulation, and two experimental storylines representing excessively favorable (high surface warming and low atmospheric warming – hiS-loA) and unfavorable (low surface warming and high atmospheric warming – loS-hiA) conditions for typhoon intensification.**





Based on this performance evaluation we select seven representative typhoon events for the subsequent PGW-based
assessment. The four events which already showed a suitable model performance in the study by Sun et al. (2024), i.e. Neoguri
(no. 9), Hagupit (no. 12), Hato (no. 22), and Usagi (no. 17), also prove to be a suitable selection for this framework, opening
the possibility to further deepen the previous investigations. To widen the spectrum of investigated events, including events of
different typhoon categories, and based on a generally low level in the combined bias of track position, minimum central
pressure and maximum wind speed with regard to both reference data sets, we additionally select Utor (no. 7), Mujigae (no.
18), and Sally (no. 3). All seven selected events show a mean bias range of $\pm$ 7.5 hPa for minimum central pressure, $\pm$ 8.5 m
s$^{-1}$ for maximum wind speed, and a deviation in track position of no more than 65 km.

The historical typhoon tracks obtained from the reference data and the ERA5-driven simulations for these seven events are
displayed in Fig. 5a. Figure 5b depicts the tracks of the ERA5-driven simulations, the 16 comprehensively perturbed PGW
simulations, and the 2 experimental storylines. Although the typhoon eye did not directly cross the PRD in all instances, the
impact on the PRD is nevertheless present by the eye-wall surrounding the storm center and the outer rain bands. Across all
simulations the agreement of the historical and simulated tracks is high. This also applies to the simulations under the purely
temperature-based PGW modulation which are not displayed in Fig. 5 The high performance of the selected spectral nudging
scheme enables the possibility to assess and compare the historical and future typhoon intensities independent of the
environment the storms passed throughout the simulations.

**3.2 PGW evaluation**

**3.2.1 Typhoon intensities of comprehensively driven PGW experiments**

The statistical distributions of minimum central pressure and maximum wind speeds under the comprehensively driven PGW
simulations for the seven selected typhoon cases are presented as boxplots in Fig. 6. The runs for all 16 GCM-based runs are
statistically compared to the reference simulation driven by un-perturbed reanalysis data. For the impact-relevant ranges of the
distributions, the maximum/minimum values of the respective variable for each of the underlying runs are additionally shown.
In terms of minimum central pressure (Fig. 6), a median reduction of minimum central pressure under future conditions occurs
for six of the seven events, varying from -2 hPa to -6 hPa. The projected decreases are significantly higher for the extreme
minimum values and reach up to -14 hPa, however the spread between the 16 GCM-based runs is high. The general indication
is therefore a slight increase in the general typhoon intensity with respect to minimum central pressure and stronger increases
in the most extreme ranges of the distribution. Four of the six events for which a median decrease is indicated, i.e., Hagupit,
Usagi, Utor, and Sally, show a decrease of minimum central pressure in all of the 16 GCM-based runs. For Mujigae, on the
other hand, the median including all future simulations is increased by 6 hPa, indicating a tendency towards a lower typhoon
intensity. However, while three of the 16 GCM-based runs indicate a distinctively lower intensity, the other models indicate
higher intensities for the extreme values.





Due to their close physical relationship, the projected changes in maximum wind speed (Fig. 6b) correlate closely with those

of minimum central pressure. A general tendency towards increased maximum wind speeds under future conditions is indicated

by the model results, but this signal is less robust than for minimum central pressure. The same six of the seven events show

an increase in the median of up to 6 m s⁻¹, while the median for Mujigae is decreased under future conditions. In terms of the

overall maximum values, maximum increases lie within a range of 2 m s⁻¹ to 11 m s⁻¹. While the majority of model runs

indicate increased wind maximums, the robustness of these increases is less dominant, and more GCM-based runs than for

minimum central pressure indicate reduced maximum values. Only Usagi indicates an increase for all 16 GCM-based runs.

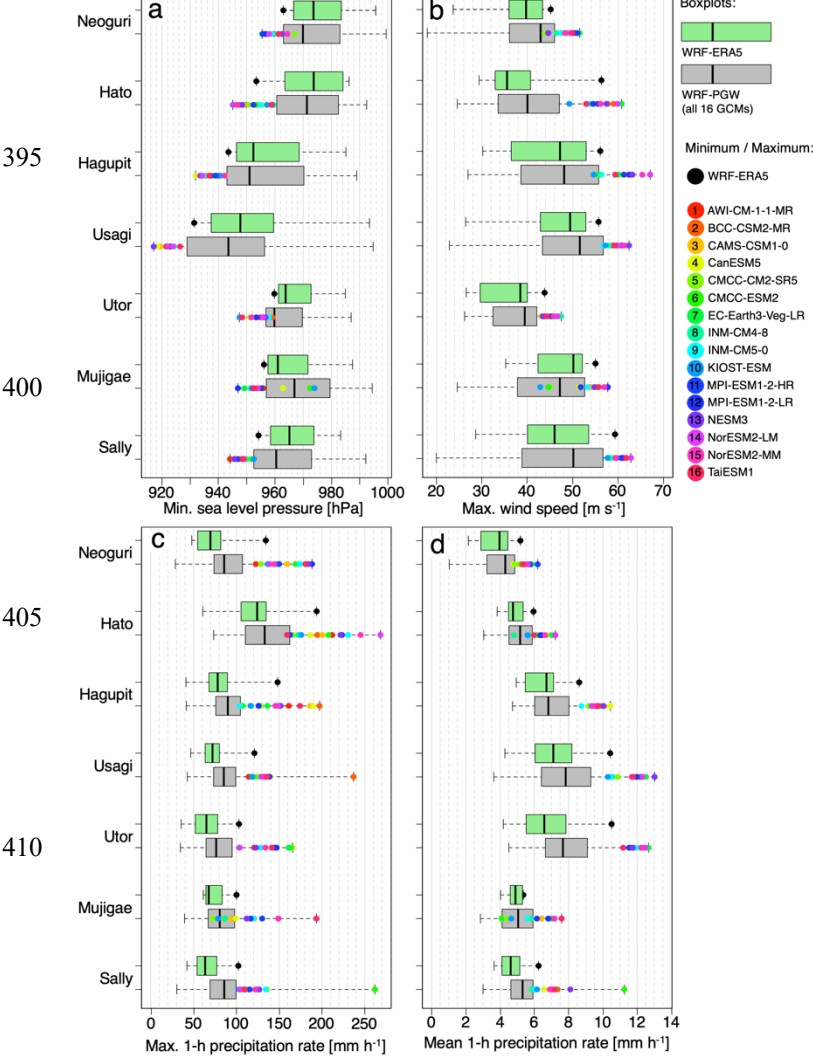

**Figure 6: Statistical distributions of comprehensively driven PGW experiments using drivers derived from 16 GCMs for seven representative typhoons. a) minimum central pressure, b) maximum wind speed, c) maximum 1-hourly precipitation rate, d) mean 1-hourly precipitation rate. Boxplots are shown for the reference simulation driven by reanalysis and all combined time series of the 16 PGW-based runs. Extreme values on high-impact end of distribution are given by circles, reference simulation in black and 16 PGW-based runs in colors.**





The projected changes in the maximum one-hourly precipitation rate are shown in Fig. 6c. All seven events show an increase in the median of all PGW-based runs of around 10 mm h$^{-1}$ to 25 mm h$^{-1}$. The potential increases within the individual storylines are significantly higher, reaching above 40 mm h$^{-1}$. The spread of these runs, however, is high. For Hato and Hagupit, the models spread widely around the reference simulation with no clear picture of a change in precipitation intensity. For the other five events the majority of GCM-based runs confirms the general tendency of increased precipitation rates. A similar picture

is given for the mean 1-hourly precipitation rates (Fig. 6d), showing median increases across all events of up to 1 mm h$^{-1}$. Substantially more individual storylines point towards increases in the mean precipitation rate, which notably exceed the increases suggested by the median and reach up to 2.5 mm h$^{-1}$.

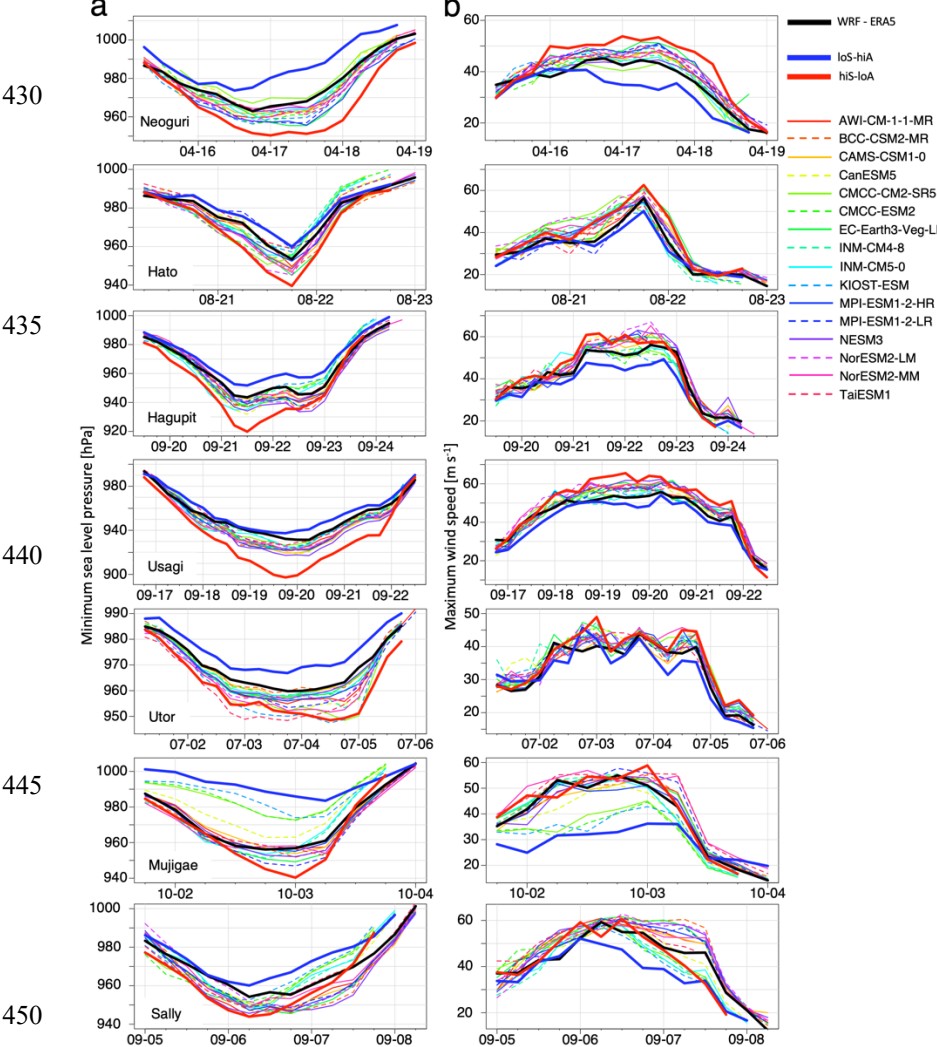

**Figure 7: Time series of a) minimum central pressure and b) maximum wind speed for the reference simulation (black) and 16 GCM-driven PGW runs (thin lines in colors), as well as 2 storylines of excessively favorable (high surface warming and low atmospheric warming, "hiS-loA", bold) and unfavorable (low surface warming and high atmospheric warming, "loS-hiA", bold) conditions for typhoon intensification. a) minimum central pressure, b) maximum wind speed.**





### 3.2.2 Typhoon intensities of thermodynamical experiments

The time series for minimum central pressure and maximum wind speed are given for the 16 GCM-based runs and the two thermodynamical PGW experiments in Figure 7. The thermodynamical PGW runs with low surface temperature warming and high atmospheric warming are labelled "loS-hiA" and the runs with high surface temperature warming and low atmospheric warming are labelled "hiS-loA". As expected, the runs with high surface temperature warming generally show higher intensities during the tracked period, often exceeding the span of the 16 GCM-based runs. Vice versa, the runs with low surface temperature warming are among the weakest realizations and often weaker than the ERA5-based reference simulation.

For minimum central pressure (Fig. 7a), the minimum values for the low surface warming experiments are higher than any other simulation for all seven events, indicating significantly reduced typhoon intensities. This increase in pressure reaches up to 10 hPa for most events and up to 45 hPa for Mujigae. The peak intensity of the high surface warming experiments is higher than all other simulations for six of the seven events. While the simulation for Utor is mostly in line with the most extreme case of the 16 GCM-based runs and the increase is minimal for Sally, the overall minimum central pressure is reduced by 7 to 20 hPa compared to the most extreme of the 16 GCM-based runs for the other five cases. While the spread between the different realizations is small at the beginning and end of the time series, it increases towards the time of maximum intensity. This applies to the 16 GCM-based simulations as well as the two experimental storylines.

For maximum wind speed (Fig. 7b), the overall picture is similar, but the differences between the experimental runs and the 16 GCM-based runs are less distinct. For most typhoons, the low surface warming experiments are less intense than all other simulations. The high surface warming experiments are mostly near the most extreme of the 16 GCM-based runs and exceed these in terms of the maximum values for Neoguri, Hato, Usagi, Utor, and Mujigae. The magnitude of this exceedance, however, is low at ca. 2 m s$^{-1}$ to 4 m s$^{-1}$. Similar to minimum central pressure, however less distinct, there is an indication of a higher model spread towards the time of peak intensity

For the maximum 1-hourly precipitation rate (Fig. 8a), as the results in general inherit a high level of variability, the drawing of conclusions is aggravated. The high surface warming experiments are mostly within the higher range of the 16 GCM-based runs, but the maximum values are often undercut. The low surface warming experiments are among the runs with the lowest precipitation intensity and often similar to the reference simulations.

The overall picture of the maximum precipitation rate is also present for the mean 1-hourly precipitation rate (Fig. 8b). Neoguri, Hagupit, Usagi, and Utor show consistently high precipitation rates for the high surface warming experiments that, in terms of maximum values, exceed all 16 GCM-based runs. While still in the upper range of all realizations, the simulated precipitation rates for the other three events are, however, exceeded by some of the 16 GCM-based runs. A similar picture is given for the low surface warming experiments. A clear reduction in the mean precipitation rate compared to the reference and 16 GCM-based runs appears for Neoguri, Hagupit, Usagi, and Sally. The runs of the other three events are closer to the reference simulation and the lower range of the 16 GCM-based runs. There is no clear difference in the model spread between the beginning, end, and peak intensity times for both precipitation metrics.



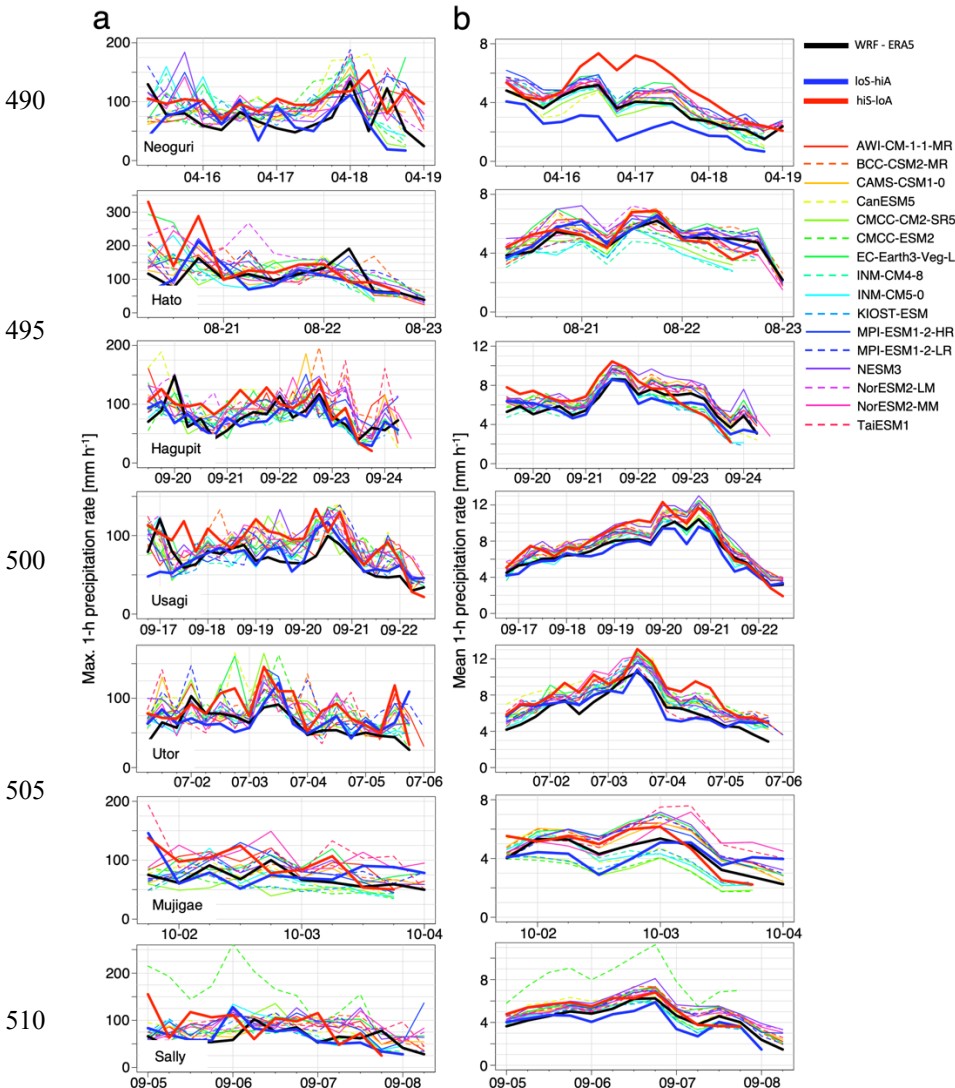

**Figure 8: same as Fig. 6, but for a) maximum 1-hourly precipitation rate, b) mean 1-hourly precipitation rate.**

### 3.2.3 Integrated kinetic energy assessment

The typhoon damage potential according to the assessment of the integrated kinetic energy (IKE) for the thermodynamically adjusted PGW runs are shown in Fig. 9. For all seven events, the IKE is larger under the high surface warming experiment. The events that were historically classified within a low typhoon category (compare Fig. 4) do not necessarily show lower IKE levels. Utor, for example, while conventionally classified as a category 1 typhoon, shows the highest IKE of all inspected events, while Usagi, historically a category 4 typhoon, shows IKE levels in the medium range. It can also be seen that the overall maximum wind speed within the entire three-dimensional field of the detected system, as shown in Fig. 9, does correlate





with the IKE, but does not reflect its variability entirely. This can be expected, since wind speed is only one of the variables contributing to the IKE.

The magnitude of the difference between the high and low surface warming experiments varies greatly among the events. The highest differences appear for Usagi, Hagupit, and Utor, at 500 MJ to 700 MJ, while the lowest differences appear for Neoguri, at 100 MJ. From a percentage perspective, the differences between the low surface warming and high surface warming

experiments are substantial for some of the inspected events. The IKE of the high surface experiments for Hagupit and Mujigae, for example, is increased by ca. 50%. For Usagi, the relative increase is ca. 35% and ca. 27% for Hato, while it is in the range of 4.5% to 8.5% for the remaining typhoons. Close to the time of peak intensity, the spread between the two experimental storylines is the largest for six of the seven events, with Hato as exception.

**Figure 9: Typhoon damage potential based on integrated kinetic energy (IKE) for 2 storylines of excessively favorable (high surface warming and low atmospheric warming, "hiS-loA") and unfavorable (low surface warming and high atmospheric warming, "loS-hiA") conditions for typhoon intensification. Overall maximum wind speeds within the 400 km x 400 km 3D field around the storm center are indicated by dashed lines.**



## 4 Discussion

### 4.1 Historical performance of simulated typhoons

When comparing our model performance with regard to the seven selected cases to that of former studies, our results are within the range of variability that was formerly deemed acceptable. The range of ± 7.5 hPa mean deviation in minimum central pressure and ± 8.5 m s$^{-1}$ for maximum wind speed, which includes all seven selected typhoons, lies close to the optimized error margins developed by Di et al. (2019) for a similar domain in the Northwest Pacific. Similar error margins were also accepted by Cha et al. (2011) and Delfino et al. (2022). There is a general tendency towards an overestimation of maximum wind speed within the selected set up which may lead to increased wind maxima, even when the minimum central pressure is accurately captured or even slightly overestimated. This was also indicated by Delfino et al. (2023), whose set up was adopted for this study. As this applies to both the reference simulations and the PGW simulations, we do not expect this overestimation to significantly impair the comparison of the simulations under historical and projected future conditions. A strength of the selected WRF set up is the close agreement of the historical and simulated tracks, which in none of the simulations of the seven selected cases deviate by more than 65 km, demonstrating the high performance of the spectral nudging technique.

It is important to point out that the choice of reference data significantly impacts the quality assessment of WRF in our case. There is no clear indication towards which reference data set our model results show more or less bias, as the preciseness of the simulations varies individually for each event. Both cases, an overestimation with reference to one, and an underestimation with reference to the other reference data set, and vice-versa, are present within our results. However, the linear relationship between minimum central pressure and maximum wind speeds is more distinct when comparing to the CMA best track data. Discrepancies between the two reference data sets were also discussed by Barcikowska et al. (2012) and Knapp and Kruk (2010). These studies found the differences to be relatively low. Although we applied the corrective factor to the JMA wind speed data, as proposed by Lee et al. (2012), the discrepancies remained in our case.

### 4.2 Typhoon intensity

All of the 18 inspected storylines indicate an increase in sea surface temperatures across all months that are relevant for this study. From this point of view, an increase in the intensity of typhoons within our domain is probable. However, this is countered by a projected increase in the thermodynamic stability of the atmosphere, caused by increased levels of warming in the mid-troposphere compared to the near-surface atmosphere. Taking both factors into account, our results show a general tendency towards a more intense unfolding of the inspected events under SSP5-8.5. This is most clearly shown by the intensification of six out of 7 studied typhoon events with respect to their median minimum pressure and wind maximum, and an intensification across all events according to their median precipitation mean and maximum from the 16 inspected storylines. This tendency and the magnitude of changes are in line with what was pointed out by former investigations, for example on a global basis by Seneviratne et al. (2021) and Knutson et al. (2020), as well as for the Northwest Pacific by Delfino et al. (2023) and Chen et al. (2020). However, as our results further demonstrate, the 16 inspected storylines do not



unanimously point in this direction and vary greatly between events. For example, 94 of the combined 96 model runs for the two most intense inspected typhoons Hagupit and Usagi show an intensity increase regarding central pressure, maximum wind speed and the mean precipitation rate. On the other hand, for Hato and Mujigae, the results are considerably more ambiguous
with storylines indicating both an increase and decrease in intensity. What can be noted, on the other hand, is that the results show a larger deviation in the extreme values when storylines indicate an increase in intensity. The projected increases in median precipitation are also notable, which in contrast to central minimum pressure and maximum wind speed are present for all seven events. In addition, the projected increases are significant and, from a relative perspective, exceed those of pressure and wind. Although an increase in thermodynamic stability is projected by all models, the projection of a general increase in
precipitation levels in a warmer atmosphere can be expected under the Clausius-Clapeyron relation (Pall et al., 2007). Yet, there remain storylines under which precipitation is decreased and this depends heavily on the selected typhoon.

The thermodynamical experiments were conducted to even better capture the physical limits of the selected framework and represent extreme scenarios of high surface and low atmospheric warming and vice-versa. For minimum central pressure, these experiments indeed showed an increased span. For the more impactful maximum wind speed, however, the experimental
simulations were often similar to the span provided by the 16 GCM-based runs, although for the majority of cases the overall maximum values within the 16 GCM-based runs were slightly exceeded by the high surface warming scenario. For both precipitation metrics the differences between the 16 GCM-based and the experimental runs were even less distinct, and the experimental runs rather reduplicated the span of the 16 GCM-based runs.

In the context of the crucial thermodynamical drivers of the PGW simulations, no distinct relationship of the magnitude of a
thermodynamical delta and the resulting typhoon intensity could be established. It could have been expectable that a storyline with an excessive increase in sea surface temperature would result in a higher typhoon intensity. In the case of the 16 GCM-based runs, this cannot be distinctively found in the results. For the two experimental storylines, there is an indication that an excessive near-surface warming leads to increased typhoon intensities and this could be expected from a physical point of view. But since some of the 16 GCM-based storylines, which we expected to be less extreme as the experimental storylines,
exceeded the range of the experimental storylines in terms of precipitation rates and maximum wind speeds, we consider variations in typhoon intensity under a PGW scheme to not solely rely on thermodynamic drivers. These variations may, for example, also occur due to changes in vertical wind shear imposed under PGW (Olschewski and Kunstmann, 2024).

The two thermodynamically-driven storylines of this study largely reflect the range of intensification suggested by the 16 GCM-based runs. Based on this, the assessment of the typhoon damage potential using IKE shows, with minimal limitations,
a range of intensification for the inspected events under a warmer climate representative that indicated by the 16 GCM-driven simulations. While the excessively favorable storyline shows typhoon intensities close to or exceeding the maximum of the 16 GCM-based runs, the unfavorable storyline shows intensities that are comparable to the reference simulation under historical climate conditions. The difference in IKE, respectively the damage potential, between the two storylines is considerable for specific events. This is especially the case for Mujigae, Hagupit, and Usagi, with the former two being the most intense
typhoons out of the seven inspected cases, purely based on maximum wind speed. Our results further demonstrate large



discrepancies between the typhoon intensities based on wind speed and based on the IKE. There has been an extensive discussion about the suitability of the Saffir-Simpson Hurricane Wind Scale as a metric for tropical cyclone intensity solely based on wind speed in the past (Jordan and Clayson, 2008; Bloemendaal et al., 2021; Walker et al., 2018; Wehner and Kossin, 2024; Kantha, 2006) Based on our findings, we recommend a more in-depth re-evaluation of typhoon intensity under multiple

sophisticated approaches of determining typhoon intensity, also considering their respective damage potential that not only depends on wind speed.

### 4.3 Limitations of the study design

There are limitations that apply to our study. Although there exist extensive investigations on our selected modeling framework (Delfino et al., 2022; Sun et al., 2024), the inclusion of a single model set up imposes a limitation on the robustness of our

results. To set the research focus on the PGW-based simulations and the provision of an extensive number of storylines, we opted for an established and previously tested model set up and made an assumption on the orthogonality of our results based on Xue et al. (2023), i.e. the variations in typhoon intensity based on the PGW scheme to behave similar for different model set ups. To increase the robustness of our results, a combined uncertainty assessment of model configurations and PGW frameworks could be conducted with increased computational capacities, using an extensive ensemble of model initializations,

parameterizations, and storylines.

The selected sample size of 7 typhoons is too small to draw a generalized conclusion on typhoon intensity over the PRD under climate change. This was not the purpose of our PGW framework, as we specifically intended to address particular historical events with an influence on the PRD and a high simulation performance. To further assess typhoon intensity on a general level, an increased number of historical events could be considered in a follow-up study, next to probability-based long-term

simulations. The limitation of reduced sample size also applies to the assessment of the strongest typhoons, for which former studies projected a distinct increase in intensity (Seneviratne et al., 2021). However, our results do confirm that the most distinct intensity increase occurs for the strongest typhoons in our sample.

### 5 Conclusions

Using a modeling framework based on Pseudo-Global Warming, we conducted simulations of historical typhoons affecting

the Pearl River Delta under modified lateral and boundary conditions derived from CMIP6. We followed a storyline approach based on Shepherd et al. (2018) to gain an in-depth view into the uncertainty range of the projected changes in typhon intensity. In addition to the 16 storylines derived from CMIP6, we created two storylines based on thermal climate change signals resembling excessively favorable and excessively unfavorable conditions for typhoon intensification. The assessment of typhoon intensity primarily focused on central minimum pressure and maximum wind speed, while additionally accounting

for track accuracy. For a deeper investigation of the simulations under future climate conditions we also investigated mean





and maximum precipitation levels as well as the integrated kinetic energy (IKE) as advanced measure for the estimation of typhoon damage potential. Concerning our research questions, we come to the following conclusions:

a) Only a slight overestimation of maximum wind speeds can be detected for the selected model framework, next to an underestimation of intensity for the strongest typhoons. The deviations for the former still lie within a range that was previously accepted within tropical cyclone research, while the deviations for the latter led to the exclusion of super typhoons Haima and Mangkhut from our further investigations. Future studies may be conducted to account for uncertainties in the model initialization and parameterization in combination with the PGW and storyline configuration.

b) The median values combining all 16 GCM-based storylines are intensified for six of the seven investigated typhoon events, excluding Mujigae. The median increases are up to 4 m s$^{-1}$ for maximum wind speed as well as 1 mm h$^{-1}$ for mean and 15 mm h$^{-1}$ for maximum precipitation rates. Median central minimum pressure is decreased by 5 hPa. However, the intensification of the extreme values of these metrics is significantly higher, resulting in reduced minimum pressures of up to 15 hPa, increased wind maxima of up to 11 m s$^{-1}$, increased mean precipitation rates by up to 2.5 mm h$^{-1}$ and increased maximum rates by up to 50 mm h$^{-1}$ in general and outliers reaching increases of more than 100 mm h$^{-1}$. We conclude most of the seven typhoons that were intensively inspected may occur with a higher intensity than historically observed. This is reflected by the projected changes in the median, but also by the majority of storylines showing lower values of central pressure, increased maximum wind speeds and higher precipitation rates.

c) For some instances the resulting intensities of the experimental storylines of excessively favorable and unfavorable conditions for typhoon intensification exceed those of the 16 GCM-based storylines, indicating that the maximum potential increase may be higher than what is suggested by the GCM-based storylines. This applies to typhoons Neoguri and Usagi in terms of minimum central pressure and maximum wind speed. In most cases, however, while the historical typhoon intensity is undercut by the unfavorable storyline, the favorable storyline is similar to the maximum of the range suggested by the GCM-based storylines. The experimental storylines therefore prove to offer a limited added value, while mostly in line of the range suggested by the GCM-based storylines.
The ensemble storyline approach proves to offer a more in-depth view into the uncertainty range of the projected increase in typhoon intensity, compared to the use of an ensemble mean for PGW. Increases exceeding those of the median, and respectively that of an ensemble mean approach, would go undetected and these simulations may not reflect the true atmospheric state under which these events may occur in the future. Treating each of the 18 storylines equally can significantly improve our understanding on the potential future unfolding of similar events and facilitate the development of protective measures. It must also be noted that there exist storylines within our framework that show a reduction of typhoon intensity compared to the historical cases, indicating that the severest occurrences may be averted when the low-impact storylines are pursued by actions of climate change mitigation.



### Data availability

The ERA5 reanalysis data used to drive the regional climate model is available upon registration from the Climate Data Store by Copernicus Climate Change Service (C3S). Links are provided by Hersbach et al. (2023a) and Hersbach et al. (2023b). The best track reference data is publicly available without registration at https://tcdata.typhoon.org.cn/en/zjljsjj.html (CMA, last accessed 07 March 2024) and https://www.jma.go.jp/jma/jma-eng/jma-center/rsmc-hp-pub-eg/besttrack.html (JMA, last accessed 07 March 2024). The WRF configuration that was used in this study is described by Delfino et al. (2023) All data used to conduct this study will be made available upon request.

### Author contributions

P.O., Q.S., J.W., Y.L., Z.T., L.S., and P.L. conceptualized the study. Data curation was carried out by P.O., Q.S., and J.W. The formal analysis was carried out by P.O., Q.S., J.W., T.C.S., B.B. and P.L. The model was set up and run by Q.S., J.W., and J.A. The statistical pre- and post-processing was conducted by P.O. Funding acquisition, project administration, and supervision were carried out by J.W., Z.T., L.S., H.K., and P.L. P.O. visualized the data and prepared the original draft. All authors provided useful feedback throughout the preparation process and review.

### Competing interests

The contact author has declared that none of the authors has any competing interests.

### Acknowledgements

We express our deep gratitude to Benjamin Fersch, Frank Neidl, and Christoph Sörgel (all KIT) for arranging the necessary computational capacities at the Linux cluster of KIT IMKIFU in Garmisch-Partenkirchen. We thank Copernicus Climate Change Service (C3S) and the European Centre for Medium-Range Weather Forecasts (ECMWF) for the public provision of ERA5 reanalysis data as well as the DKRZ (German Climate Computing Center) and the ESGF (Earth System Grid Federation) for the public provision of CMIP6. We also thank the China Meteorological Administration (CMA) and the Regional Specialized Meteorological Centre (RSMC) Tokyo for the public provision of the two tropical cyclone best track data sets.





**Financial support**

This study was conducted in the framework of the Sino-German project Mitigating the Risk of Compound Extreme Flooding Events, *MitRiskFlood*, jointly funded by MOST (grant no. 2019YFE0124800) and the German Ministry of Education and Research (BMBF, grant no. 01LP2005A). Q.S. and Y.L. were financially supported by the Chinese Scholarship Council (CSC).

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
