# Peer review of "A storyline-based approach towards changing typhoon intensities over the Pearl River Delta under future conditions using Pseudo-Global Warming"

_Hydrology and Earth System Sciences, 2024_

## Community Comment (CC1)

**Review of "A storyline-based approach towards changing typhoon intensities over the Pearl River Delta under future conditions using Pseudo-Global Warming" by Olschewski et al.**

The authors presented a thorough storyline-based analysis over the Pearl River Delta under future conditions using Pseudo-Global Warming (PGW) modeling techniques. They perturbed the historical initial and boundary conditions obtained from ERA5 reanalysis and handed to the regional model WRF according to the climate change signal projected by the CMIP6 ensemble under SSP5-8.5, in which each of the 16 selected CMIP6 models acted as the basis for an individual storyline of future typhoon intensity. Results indicate a general increase in typhoon intensity across all metrics for six of the seven inspected typhoons. This increase is notably higher for specific storylines, and the projected increase in the extreme values of the inspected metrics significantly exceed the median change of all storylines.

Overall, the manuscript is well-written and well-organized, and is equipped with a large workload and high-quality numerical experiments as key supporting materials. I believe the method is scientifically valid and the results are logical and reasonable. Based on the current phase of the manuscript, if the authors can properly address the following comments, I would recommend accepting it for publication.

**Minor Comments**

1. The authors utilized CMA and JMA best track data to evaluate the model's performance. Meanwhile, the Hong Kong Observatory (HKO) also provides best track data for tropical cyclones that affect the PRD. Due to Hong Kong's geographical location, HKO's best track data is generally considered to be of high quality for the PRD region. If it does not require a significant amount of extra work, the authors may want to consider comparing their results with HKO's best data in Figures 4 and 5.

2. The CMA best track archives the 2-minute average maximum sustained wind speed at the 10-meter level, while JMA archives 10-minute values. On the other hand, the default output for WRF is the instantaneous wind speed of the grid area mean, which may be related to the integration time step and grid area. It would be helpful if the authors could clarify how they compared these different definitions of wind speed, and whether they used any conversion coefficients.

3. The authors employed spectral nudging to enhance the performance of TC tracks. Although the authors have justified that the nudging operation does not affect the inner core of the simulated typhoon, studies have shown that nudging in perturbed experiments may limit the intensity of TCs (Moon et al. 2018; Li et al. 2024). Therefore, it is a trade-off to turn on nudging to reproduce comparable TC tracks in the PGW experiments. The authors may want to consider discussing this in L250-255.

4. L165: Please give specific reasons for the preference of using grid-cell basis compared to spatial mean.

5. L175-180: Although seven selected typhoon events did not occur in May and June, it is still valuable to fill the 2-month gap in Figure 2 to give a complete picture of summer-time seasonal cycle of global warming signal for readers' reference.

6. L210: The authors have provided a justification for using WRF instead of HWRF. However, it is important to note that the ocean coupling effect, such as cold wake feedback and wind-sea wave interactions, may also impact the intensity of TCs, which cannot be captured by WRF-only simulations (e.g., Mogensen et al. 2017; Magnusson et al. 2019; Li et al. 2022). Therefore, the authors may want to address some of these limitations in their study.

7. L230-235: The authors utilized the Rapid Radiative Transfer Model (RRTM) for longwave radiation and the Dudhia scheme for shortwave radiation in their WRF configuration. To the best of my knowledge, RRTM shortwave scheme is typically used in conjunction with the longwave radiation scheme. Therefore, it would be

beneficial if the authors could provide a justification for why they did not use the RRTM shortwave scheme in their study.

**References**

Li, Z., Tam, C.Y., Li, Y., Lau, N.C., Chen, J., Chan, S.T., Dickson Lau, D.S. and Huang, Y., 2022. How Does Air‑Sea Wave Interaction Affect Tropical Cyclone Intensity? An Atmosphere‑Wave‑Ocean Coupled Model Study Based on Super Typhoon Mangkhut (2018). Earth and Space Science, 9(3), p.e2021EA002136.

Li, Z., Fung, J.C., Wong, M.F., Lin, S., Cai, F., Lai, W. and Lau, A.K., 2024. Future changes in intense tropical cyclone hazards in the Pearl River Delta region: an air-wave-ocean coupled model study. Natural Hazards, pp.1-16.

Magnusson, L., Bidlot, J.R., Bonavita, M., Brown, A.R., Browne, P.A., De Chiara, G., Dahoui, M., Lang, S.T.K., McNally, T., Mogensen, K.S. and Pappenberger, F., 2019. ECMWF activities for improved hurricane forecasts. Bulletin of the American Meteorological Society, 100(3), pp.445-458.

Mogensen, K.S., Magnusson, L. and Bidlot, J.R., 2017. Tropical cyclone sensitivity to ocean coupling in the E CMWF coupled model. Journal of Geophysical Research: Oceans, 122(5), pp.4392-4412.

Moon, J., Cha, D.H., Lee, M. and Kim, J., 2018. Impact of spectral nudging on real‑time tropical cyclone forecast. Journal of Geophysical Research: Atmospheres, 123(22), pp.12-647.

---

## Author Comment (AC1)

**Response to Reviewer 1 (RC1):**

**Comment:** This paper presents numerical simulations for a series of typhoon events using the WRF model and PGW methods to extract the SSP585 far future scenario global warming signal from 16 CMIP6 models, which is added to the ERA5-based historical climate as the IC BC of the model. The authors have invested significant effort in model simulation and demonstrated methodological rigor; the article is well-written and easy to read. It highlights changes in intensity and precipitation in seven typhoon cases under future scenarios, compared to historical control runs, as well as the impact of varying surface and low-level atmospheric temperatures on typhoon intensity.
**Reply:** We thank Reviewer 1 for the positive feedbacks.

The topic is not related to hydrology or urban hydrology, I'm not sure if it coincides with the scope of HESS journal.
**Reply:** We believe that our topic is closely related to hydrology, particularly urban hydrology. As mentioned by Reviewer 1 in a previous comment,
- Our study addresses an important hydrological research question: "How will extreme weather and climate events change in a warming climate, especially concerning intensifying landfalling typhoons?";
- We focus on evaluating and understanding the changes in typhoon intensity induced by warming, specifically those making landfall in the Pearl River Delta, where are established metropolises and rapidly growing megacities with a population of approximately 86 million;
- Our findings have important implications for enhancing follow-up studies from an impact analysis perspective, such as assessing compound flood hazards from rainfall and storm surge.

We have carefully checked the Aims & scope of HESS journal by following the link https://www.hydrology-and-earth-system-sciences.net/about/aims_and_scope.html. We believe that our study is within the scope of HESS journal, since "*HESS encourages and supports fundamental and applied research that advances the understanding of hydrological systems, their role in providing water for ecosystems and society, and the role of the water cycle in the functioning of the Earth system.*"

Our study coincides with the second scope of HESS "*2. the study of the spatial and temporal characteristics of the global water resources (solid, liquid, and vapour) and related budgets, in all compartments of the Earth system (atmosphere, oceans, estuaries, rivers, lakes, and land masses), including water stocks, residence times, interfacial fluxes, and the pathways between various compartments;*"

**Comment:** There are some merits in the study, but there are substantial flaws that prevent the manuscript from publication. The discussion lacks depth concerning the physical mechanisms of climate change impacts on typhoons, especially for the typhoon dynamic and thermodynamic changes. The paper primarily presents changes in typhoon intensity and precipitation as projected by the WRF model and PGW method under a warmer climate, but it fails to systematically explore the underlying dynamic and thermodynamic reasons for these changes. This omission limits the depth of understanding of the complex physical processes involved. Moreover, the conclusions presented are not particularly novel and innovative, focusing mainly on the increase in typhoon intensity and precipitation due to global warming. Addressing these issues could significantly improve the paper's quality.
**Reply:** We thank the Reviewer for the feedback.

In this revision we have addressed your concerns as follows:
The underlying physical mechanisms driving warming-induced changes in typhoon intensity and precipitation are explored in more detail. Specifically, we have closely followed the Reviewer's suggestions to shed light on the dynamic and thermodynamic factors behind these changes. For example, we compare convective available potential energy (CAPE), vertical winds, and relative

vorticity from PGW simulations with those from ERA5 simulations to improve our understanding of the complex physical processes involved.

The Conclusions section is thoughtfully revised to highlight our key scientific findings, specifically, physical mechanisms driving changes in typhoon intensity and precipitation under varying warming conditions.

**Comment:** The introductory section is cluttered and lacks a logical structure, and it is more of a thesis introduction than that of a research article. Furthermore, while the authors spend considerable time discussing existing research gaps in current PGW studies, they barely touch upon the specific effects of climate change on typhoon intensity and the physical mechanisms behind it. The authors mention previous studies but do not systematically explain the results and progress made in this field. For instance, whether climate change will lead to stronger typhoons in the western Pacific, how much will the typhoon intensify with one degree SST warming, and how climate change can affect typhoons' intensity and precipitation. There should already be plenty of research covering these topics. Strengthening the introduction would make it easier for readers to grasp what has been accomplished and the scientific basis of it.
**Reply:** Our introductory section is designed to present the pseudo global warming approach and the storyline approach in the context of warming-induced changes in typhoon intensity.

In this revision a comprehensive literature review has been conducted to systematically present the background and current state-of-the-art research on warming-induced changes in typhoon intensity. Our revised introductory section specifically highlights key findings and achievements in understanding the plausible physical mechanisms of typhoons under warming conditions, including atmospheric effects, oceanic influences, and water cycle dynamics. This strengthens the introduction section and improve the readability.

**Comment:** On page 5, line 155, the manuscript states that 28 typhoons were initially selected for performance assessment, but only seven were ultimately chosen for detailed study. This selection process raises questions about the model's ability to accurately simulate typhoon tracks and intensities. As the author mentioned in the introduction: "How well are historical typhoons affecting the PRD represented by our selected model set up regarding track accuracy and intensity?". If only seven out of 28 typhoons demonstrate reasonable model performance, this suggests potential limitations in the model's reliability for simulating such events.
**Reply:** We acknowledge the limitations in our current model setup for simulating typhoons. We are fully aware of the uncertainties arising from model physical schemes, configurations, synoptic conditions, and warming scenarios. However, before reaching the final model setup (shown in Figure 4), we conducted numerous preliminary simulations with different domain sizes, domain numbers, grid spacing, and physical schemes to identify a suitable model setup. These preliminary simulation results indicated that it is challenging to find a single model setup that performs well for all 28 typhoons. As a result, our PGW analyses are limited to the 7 selected typhoons, where the WRF model setup led to good performance.

In this revision additional PGW simulations for one of the typhoons with poorer performance are conducted to assess whether the performance of the WRF model impacts the credibility of the remaining PGW simulations. The limitations of WRF in simulating typhoons in the region are critically elaborated in the "4.3 Limitations of the study design" section.

**Comment:** On page 8, line 250, the use of spectral nudging is mentioned, but it's unclear whether it was applied across all domains. Typically, spectral nudging is used at the outermost model domain to minimize its impact on simulation results. Clarification of how spectral nudging was implemented in this study is needed for a better understanding of its influence on the results.
**Reply:** The clarification of using spectral nudging has been added into the Methods section.

"In our study, spectral nudging is applied in both the outermost and innermost model domains to prevent the model from deviating from its boundary conditions. We used spectral nudging only for horizontal wind components (i.e., $u$ and $v$) above the 500 hPa level."

**Comment:** The analysis of Figure 4 focuses only on the mean bias of maximum wind speed and minimum sea level pressure, which may not fully capture the model's reliability and accuracy (results are also very sensitive to the time period chosen for time averaging). A more robust approach would be to compare the time series of the model's maximum wind speeds and minimum sea level pressures against best track data, to assess both intensity and temporal accuracy more comprehensively.

**Reply:** In Figure 4, the mean bias is used to quantitively assess the model's performance in simulating typhoon intensity. However, we acknowledge that mean bias does not capture the model's performance in terms of temporal evolution.

In this revised version we have introduced additional metrics, such as the Kling-Gupta efficiency (KGE), to more comprehensively evaluate temporal accuracy.

**References:**

Gupta, H. V., Kling, H., Yilmaz, K. K., & Martinez, G. F. (2009). Decomposition of the mean squared error and NSE performance criteria: Implications for improving hydrological modelling. Journal of Hydrology, 377, 80–91. https://doi.org/10.1016/j.jhydrol.2009.08.003

Kling, H., Fuchs, M., & Paulin, M. (2012). Runoff conditions in the upper Danube basin under an ensemble of climate change scenarios. Journal of Hydrology, 424–425, 264–277. https://doi.org/10.1016/j.jhydrol. 2012.01.011

**Comment:** Section 3.2.2 discusses experiments comparing loS-hiA (low surface - high atmospheric temperatures) and hiS-loA (high surface - low atmospheric temperatures) scenarios. While the comparison is interesting, the deviation from realistic conditions and the lack of in-depth analysis on the physical mechanisms influencing typhoon dynamics is a major drawback. The authors only describe time series plots of typhoon intensity and precipitation, which is seriously lacking in the analysis and explanation of the mechanism from the point of view of typhoon dynamics, thermodynamics, and physics. A reasonable research article should not only pose scientific questions to identify problems but also provide well-founded explanations and reasons, rather than merely describing the observed model outputs.

**Reply:** We thank the Reviewer for guiding us toward a more in-depth analysis of the physical mechanisms.

We would like to emphasize the scientific contributions of the loS-hiA and hiS-loA scenarios. Developing these two additional storylines from available GCMs allows us (1) to quantitively assess changes in uncertainties associated with simulated hydrometeorological variables (e.g., Figures 7 and 8); (2) to enhance risk awareness by investigating deep uncertainties beyond the recognized limitations of GCM ensembles, particularly in relation to "Gray Rhino" or "Black Swan" events in the context of weather and climate extremes. In the revised version, we will include the above statements to highlight the novelty of our study.

In the revised version, the physical mechanisms behind the warming-induced changes in typhoon intensity and precipitation are systematically explored. Specifically, we closely follow the Reviewer's suggestions to shed light on the dynamic and thermodynamic factors driving these changes. For instance, we compare convective available potential energy (CAPE), vertical winds, and relative vorticity derived from PGW simulations with those from ERA5 simulations to enhance our understanding of the complex physical processes involved.

We thoughtfully revise the Conclusions section to emphasize our key scientific findings, particularly the physical mechanisms behind changes in typhoon intensity and precipitation under different warming scenarios.

**Comment:** Following up on the previous comment, after reviewing sections 3 and 4, I am concerned that the results presented in this article do not sufficiently support the research paper. While it is evident that the authors conducted a large number of experiments using the WRF model, the results section primarily showcases time series plots of typhoon intensities and precipitation (including the Integrated Kinetic Energy) from these experiments. These findings are not particularly novel and innovative, as several studies have already employed the WRF and PGW methods to investigate the impact of climate change on typhoon intensity and precipitation, often arriving at similar conclusions. My suggestion is for the authors to conduct a comprehensive analysis of the model results, comparing the simulation outcomes of PGW signals derived from individual CMIP6 models. It is helpful to determine whether the variations between different experiments stem from discrepancies in CMIP6 model global warming signals and to assess how these differences specifically affect typhoon intensity. The authors should also delve into the reasons behind changes in typhoon intensity and precipitation, detailing the specific dynamics and thermodynamics of typhoons and how these alterations influence typhoon behavior, which is a crucial aspect of this research. For example, in comparisons such as loS-hiA and hiS-loA, the study should explore how varying temperatures introduced through PGW modify the typhoon structure and dynamics, rather than merely describing the time series of typhoon intensity.

**Reply:** Thank you for your thoughtful and detailed feedback.

In this revision, we have closely followed the reviewer's suggestions. We dive into the analysis to explore the physical mechanisms behind our results. For example, we compare the WRF-PGW simulations derived from individual CMIP6 models. This comparison helps us determine whether the variations between different experiments arise from discrepancies in the global warming signals of the CMIP6 models and assess how these differences specifically influence typhoon intensity.

In this revision the physical mechanisms driving the warming-induced changes in typhoon intensity and precipitation are systematically investigated. We closely follow the Reviewer's suggestions to shed light on dynamic and thermodynamic factors. For example, we compare convective available potential energy (CAPE), vertical winds, and relative vorticity from the PGW simulations (including the loS-hiA and hiS-loA scenarios) with those from ERA5 simulations. This enables us to quantitatively examine how temperature variations introduced through PGW affect typhoon structure and dynamics.

**Comment:** As noted in Section 4.3, line 636, the author acknowledges that "The selected sample size of 7 typhoons is too small to draw a generalized conclusion on typhoon intensity over the PRD under climate change." If the model accuracy can't be improved, I will recommend that the authors extend their analysis to include all 28 typhoon cases, regardless of whether the WRF accurately reproduces the typhoon's path and intensity very accurately. This broader analysis could enable the authors to draw more general and comprehensive conclusions. For instance, they could determine how much warming per degree of SST might lead to an enhancement of typhoon intensity; and whether the increase in typhoon rainfall due to SST, atmospheric temperature, and typhoon intensity, following the Clausius-Clapeyron (CC) relationship. The authors should also compare these findings with those from existing references to provide a more robust evaluation of the impacts of climate change on typhoons.

**Reply:** We appreciate the Reviewer's suggestion to include all 28 typhoon cases, regardless of the WRF model performance. However, we have chosen to focus on the 7 selected typhoons for our PGW investigation. This choice is based on our evaluation of the model's performance using objective metrics. Since our study specifically aims to investigate warming-induced changes in typhoon intensity, these 7 selected cases represent a range of intensities from Category 1 to Category 4. For each typhoon, we conducted 18 warming scenario simulations. We believe that this sample size is sufficient to support our conclusions. Of course, we are fully aware of the uncertainties arising from

model physical schemes, configurations, synoptic conditions, and warming scenarios. However, our preliminary simulations indicated that it is challenging to find a single model setup that performs well across all 28 typhoons. Conducting WRF simulations for all 28 typhoon cases across 18 storylines would require significantly high computational resources and storage capacity. Therefore, our selection is a balance between computational efficiency and model accuracy, and our PGW analyses are restricted to the 7 selected typhoons.

In this revision we have closely followed the Reviewer's suggestions to investigate the Clausius-Clapeyron relationship. Specifically, the relationships between changes in hydrometeorological variables (e.g., precipitation or the vertically integrated atmospheric total water) are analyzed for each storm and each GCM, as well as changes in the temperature. This analysis enhances our understanding of the Clausius-Clapeyron relationship, particularly for typhoon cases under warming conditions. We focus on the uncertainties in the Clausius-Clapeyron relationship derived from the PGW simulations, for example, comparing the simulated responses of Hagupit and Neoguri forced by the CanESM5 model. Accordingly, the relevant physical mechanisms behind the increase in the impact and the spread across models are revealed. Eventually, these insights enable us to address the research question: "Do warmer models lead to more changes?".

In this revision the comparison of our findings with those from existing references is added into the "Discussion" section to provide a more robust evaluation of the impacts of climate change on typhoons.

---

## Author Comment (AC2)

**Response to Reviewer 2**

**Comment:** Olschewski et al., investigate tropical cyclones (TCs) over the Pearl River Delta and their physical changes under climate change using the storylines approach. They select historical TCs and change their initial and boundary conditions using the Pseudo-Global Warming method. The method used in this study is interesting, as it takes advantage of the flexibility that storylines offer to carry out a thorough analysis, going beyond mean/median changes in projections. I particularly liked the extra 2 storylines exploring favorable and unfavorable thermodynamic conditions, as this type of thinking helps overcome potential limitations on climate models. The different results obtained for the thermodynamic storylines support the added value of exploring them. I also like the inclusion of integrated kinetic energy (IKE) as an additional metric aimed at offering some proxy for impact. I believe the manuscript contributes to the field and is compatible with this journal. Having said that, I have some comments and questions that could provide more context to the work.

**Reply:** We are pleased that the reviewer acknowledges the significance of our work. We appreciate the constructive comments for enhancing the quality of our manuscript.

**Spectral nudging:**

**Comment:** In the methods section (line 252), spectral nudging is used to minimize spatial variability. I found the idea of combining both approaches interesting. Did you run/check simulations without applying spectral nudging? I wonder how influential this step is in your setup. I think this merits more discussion on the implications of using and not using the spectral nudging.

**Reply:** We did conduct a substantial number of preliminary simulations to evaluate the performance of our model setup, both with and without spectral nudging. Our findings indicate that the correct application of spectral nudging is crucial for accurately reproducing observed typhoon tracks. This finding aligns with our recent publication (Sun et al., 2024) and is supported by previous studies (e.g., Moon et al., 2018).

The discussions on the implications of using versus not using spectral nudging, in particular in conjunction with the pseudo global warming approach are added into the revised version.

**Reference:**

Sun, Q., Olschewski, P., Wei, J., Tian, Z., Sun, L., Kunstmann, H., and Laux, P., 2024: Key ingredients in regional climate modelling for improving the representation of typhoon tracks and intensities, Hydrol. Earth Syst. Sci., 28, 761–780, https://doi.org/10.5194/hess-28-761-2024

Moon, J., Cha, D.H., Lee, M. and Kim, J., 2018: Impact of spectral nudging on real- time tropical cyclone forecast. Journal of Geophysical Research: Atmospheres, 123(22), pp.12-647.

**Comment:** While it makes sense to remove spatial variability to make comparisons more straightforward, a spatial change in TC track as consequence of climate change would be an interesting finding and relevant for impacts. If there are results available for the runs without the spectral nudging, you could consider analyzing them as well.

**Reply:** We generally agree with the importance of quantifying spatial changes in typhoon tracks due to global warming. However, we found that global warming does not lead to noticeable changes in the simulated typhoon tracks, primarily due to the application of spectral nudging.

In the revision the spatial changes in the simulated typhoon tracks as a consequence of climate change, particularly in runs without spectral nudging are quantitively analyzed. Additionally, the uncertainties associated with the results derived from applying spectral nudging are discussed as well.

**Comment:** Since you included spectral nudging in addition to PGW, you could discuss the role (e.g., advantages, drawbacks, and when to use each one) of each method - PGW, spectral nudging, and their combined use - in simulating TCs and their use with storylines.

**Reply:** We thank the Reviewer for guiding the discussions.

The discussion regarding the role of the three methods we employed in simulating tropical cyclones and their associated storylines are enhanced. We have closely followed the Reviewer's suggestions to examine the advantages, limitations, and applicability of PGW, spectral nudging, and their combined use, particularly in the context of evaluating weather and climate extremes induced by global warming.

**Temperature levels:**
**Comment:** In figure 2 you demonstrate the resulting delta change in temperature for each model from April to October. I am intrigued to understand the direct relation between the temperature change per model for each storm and the change in relevant metrics, such as precipitation. For instance, in Figure 2b CanESM5 (yellow) shows the largest delta change for 3 months, which made me expect to see the same model causing the largest changes for precipitation across all storms. Instead, only for one case, Hagupit, we see that. Could it be that the other storms occur in months where CanESM5 is not as intense as other models? I believe adding an extra (SI) Figure comparing some results (maybe mean precipitation) to the increase in temperature per model could offer extra insights into the mechanisms behind the increase in impact and the spread across models (do warmer models lead to more changes?).
**Reply:** We thank the Reviewer for the valuable suggestions.

In this revision the relationships between changes in the hydrometeorological variables (e.g., precipitation or the vertically integrated atmospheric total water) are derived for each storm and each GCM in relation to temperature changes. This analysis enhances our understanding of the Clausius-Clapeyron relationship, particularly for typhoon cases under global warming. As suggested, we focus on the uncertainties in the Clausius-Clapeyron relationship derived from the PGW simulations, for example, comparing the simulated responses of Hagupit and Neoguri forced by the CanESM5 model. Accordingly, the relevant physical mechanisms behind the increase in the impact and the spread across models are revealed. Eventually, these insights enable us to address the research question: "Do warmer models lead to more changes?".

**Contextualization:**
**Comment:** I see that the focus of the discussion section was in comparing the results for TC estimations in the DPR with other studies. However, I miss some discussion on your approach and results from a storylines perspective. There are other storyline works that explored TCs in DPR (Qiu et al., 2022), TCs using spectral nudging (Goulart et al., 2024) or even using a similar Pseudo-Global Warming method (Dullaart et al., 2024). I believe also framing the study within the storylines field of work can provide a better contextualization and visibility to your work.
**Reply:** We thank the Reviewer for guiding the discussions.

In the revision we elaborate on the discussions on the role of the three methods we employed in simulating tropical cyclones. Specifically, the advantages, limitations, and applicability of PGW, spectral nudging, and their combined use, are discussed particularly in the context of evaluating weather and climate extremes induced by global warming. Additionally, our study is framed within the storylines field of work to enhance contextualization and increase visibility.

**Minor comments:**
**Comment:** In general, the writing is a bit long and complex, some more direct text could make the flow of the manuscript better.
**Reply:** We thank the Reviewer's feedback. Our revised manuscript is presented in a clear organizational structure.

**Comment:** Line 41: biggest natural risk factors -> I think biggest is not the best adjective here.
**Reply:** "Biggest" has been changed to "destructive".

**Comment:** Line 120: "This is based on the findings of Shen et al. (2000), Hill and Lackmann (2011), and Tuleya et al. (2016) who found that an increase in thermodynamic atmospheric stability and

increased sea surface temperatures counteract with regards to typhoon intensity" – The text is a bit confusing, and the sentence ends with a comma.
**Reply:** We have rephrased this sentence for clarity.

"The reassembling strategy is due to the fact that the warming-enhanced atmospheric stability and the increased sea surface temperature have opposite effects on typhoon intensity (Shen et al., 2000; Hill and Lackmann, 2011; Tuleya et al., 2016)."

**Comment:** Line 164: "initial and boundary conditions from ERA5 are applied on a grid-cell basis" – does it mean for each grid cell you apply a specific delta factor? So, it means that the delta factor is calculated for every grid cell and every timestep of the simulation?
**Reply:** In this study, the delta factor is calculated for every grid cell on each pressure level of the selected GCMs, but not for every timestep of the simulation.

We derive the delta factor by comparing GCM projections during the far-future period of 2071-2100 and the corresponding GCM simulations during the historical (baseline) period of 1985-2014. Temporally, the delta factor is calculated on a monthly scale, averaged over 30 years. Spatially, it is calculated for each grid cell on each pressure level of the GCMs.

To incorporate climate change signals from GCMs into the ERA5 reanalysis, we first remap the GCM-derived delta factor fields onto the grid cells of ERA5. Then, the remapped delta factor fields for a given month are added to the 6-hourly ERA5 reanalysis for that same month, facilitating pseudo global warming simulations.

In this revision the above-mentioned details about the procedure of deriving and applying delta factors have been added into the "Data and Methods" section.

**Comment:** Line 366: "simulations the agreement of the historical and simulated tracks is high" -> Is this already including the spectral nudging? If so, it could be a bit clearer.
**Reply:** Yes, the spectral nudging has been included in the simulations.

We have included this information, and the updated sentence is as follows:
"Across all simulations, the agreement between the historical and simulated tracks is high, primarily due to the application of spectral nudging."

**Comment:** Line 425: I think it could be rewritten to make it clearer and more direct.
**Reply:** As suggested, we have rewritten this sentence for clarity.

The updated sentence is as follows:
"Figure 6d illustrates that warming will increase the median of the mean 1-hourly precipitation across the seven investigated typhoons by up to 1 mm h$^{-1}$."

**Comment:** Line 624: Missing punctuation in "; Kantha, 2006) Based on"
**Reply:** A full stop has been added in the revised version.

**Comment:** Limitations (line 628): Since you already aimed for some impact proxy using the IKE, you could mention that other impact approaches (metrics, models etc.) can also enhance the study from an impact perspective, such as wind and flood modelling.
**Reply:** We thank the Reviewer for guiding the discussions.

In our revision, in addition to the IKE, the discussion on other impact analysis approaches is expanded. For example, compound events analysis using copulas (e.g., Bevacqua et al., 2017) and storm surge and inundation modeling using hydrodynamic models (e.g., Xu et al., 2024) are discussed in the revised "Limitations of the study design" section.

**References:**

Bevacqua, E., Maraun, D., Hobæk Haff, I., Widmann, M., and Vrac, M., 2017: Multivariate statistical modelling of compound events via pair-copula constructions: analysis of floods in Ravenna (Italy), Hydrol. Earth Syst. Sci., 21, 2701–2723, https://doi.org/10.5194/hess-21-2701-2017

Xu, H., Ragno, E., Jonkman, S. N., Wang, J., Bricker, J. D., Tian, Z., and Sun, L., 2024: Combining statistical and hydrodynamic models to assess compound flood hazards from rainfall and storm surge: a case study of Shanghai, Hydrol. Earth Syst. Sci., 2024, 28, 3919–3930, https://doi.org/10.5194/hess-28-3919-2024

**References:**

Qiu, J., Liu, B., Yang, F., Wang, X., and He, X.: Quantitative Stress Test of Compound Coastal-Fluvial Floods in China's Pearl River Delta, Earth's Future, 10, e2021EF002638, https://doi.org/10.1029/2021EF002638, 2022.

Goulart, H. M. D., Benito Lazaro, I., van Garderen, L., van der Wiel, K., Le Bars, D., Koks, E., and van den Hurk, B.: Compound flood impacts from Hurricane Sandy on New York City in climate-driven storylines, Nat. Hazards Earth Syst. Sci., 24, 29–45, https://doi.org/10.5194/nhess-24-29-2024, 2024.

Dullaart, Job CM, et al. "Improving our understanding of future tropical cyclone intensities in the Caribbean using a high-resolution regional climate model." Scientific Reports 14.1 (2024): 6108.

**Reply:** The recommended literatures are cited in the revised version.

---

## Author Comment (AC3)

**Response to Review 3 (RC3):**

**Comment:** In this manuscript, the authors run a set of pseudo-global warming (PGW) simulations to investigate how 7 tropical cyclones (TCs) in the Pearl River Delta (PRD) region might change under warming perturbations from 16 different CMIP6 models (plus two additional "extreme" perturbations). They report results regarding storm pressure, wind, precipitation, and integrated kinetic energy. They find that storms generally increase in intensity across all metrics for 6 of the 7 storms. The application of climate storylines has become popular in recent years. The PGW framework is one way such storylines can be performed that targets regional scales. I agree that there is utility in such simulations, particularly around understanding how climate may change in these regions and communicating these to downstream individuals interested in climate data (e.g., planners, emergency managers, etc.).

**Reply:** We are pleased that the scientific contributions and implications of our study are well-recognized.

**Comment:** TCs and climate are important topics, but there are opportunities to deepen the analysis, better interpret the science, and make the findings more impactful. As is, I find the paper somewhat uninspiring. The actual analysis of the storms themselves is fairly shallow, with only basic qualitative comparative evaluation and not providing a deeper interpretation of the dynamical structure (e.g., how the changes in stability impact the storm's axisymmetric and asymmetric circulations). Even the evaluation of aspects such as the radius of maximum wind and precipitation distributions could be interesting. Conversely, the sample of 7 TCs doesn't feel large enough to draw any meaningful conclusions and the authors do not provide any real concrete reasons for their decision to eschew a larger sample size (28 storms). Of note, there is a lack of statistical tests to evaluate if any of the changes are robust. The general findings mainly serve as further confirmation of the numerous PGW/TC simulations published over the past 10 years or so.

**Reply:** We thank the Reviewer for encouraging us to enhance the quality of our study further.

Regarding the novelty of our study, we believe that combining the storyline approach with the pseudo-global warming approach enhances our understanding of uncertainties, particularly, in partitioning the uncertainties related to the physical aspects of global warming. Specifically, we have developed two additional storylines based on thermodynamic drivers to create scenarios with excessively *favorable and unfavorable conditions* for typhoon intensification extending beyond traditional GCM-driven simulations. The development of these two additional storylines allows us (1) to quantitatively understand the changes in associated uncertainties in simulated hydrometeorological variables (e.g., Figures 7 and 8) and (2) to enhance risk awareness by investigating deep uncertainties that go beyond the recognized limitations of GCM ensembles, particularly concerning the so-called "Gray Rhino" or "Black Swan" events in the context of weather and climate extremes. These statements are included to highlight the novelty of our study.

Regarding in-depth analysis, we closely follow the Review's suggestions to examine how changes in stability impact the storm's axisymmetric and asymmetric circulation. For this analysis, we exemplarily compare the vorticity fields from PGW-driven simulations with those from ERA5-driven simulations. Additionally, the temperature-precipitation relationship is investigated to better understand the Clausius-Clapeyron relation, particularly for typhoon cases under warming conditions.

The selection of the seven typhoons for the PGW investigation is based on the model performance evaluation results, which were assessed using objective metrics to reproduce the observed maximum wind speed and minimum sea level pressure. Since our study specifically focuses on warming-induced changes in simulated typhoon intensity, the selected typhoons encompass the full range from Category 1 to Category 4. For each typhoon, we have conducted 18 warming scenario simulations, and we believe that this sample size is sufficient to conclude each typhoon (shown in Figure 6). Of course, we are fully aware of the uncertainties in the simulations stemming from model physical schemes, configurations, synoptic situations, and warming scenarios. However, our preliminary

simulations indicated that it is challenging to find a single model setup that performs well across all 28 typhoons. Therefore, our PGW analyses are restricted to the seven selected typhoons.

The above descriptions about the selection process for these seven typhoons are added into the revised version.

Regarding statistical tests, the non-parametric Mann-Whitney U-test is applied for quantitative significance testing, since the Gaussian distribution could not be assumed for all results in our study.

**Comment:** The paper reads a bit like using a tool (PGW simulations) to find a nail (TCs in the PRD). It currently is written in a way that reads very linearly and more like a technical document than a scientific paper. As is, I think the paper needs a fairly large overhaul before it can be considered for formal publication in a journal. The good news is I think this can be achieved with a better interpretation and deeper understanding of the data as opposed to additional model simulations. My major comments are described below and I think need to be critically addressed before any publication.
**Reply:** In this study, we employed a storyline approach to shed light on the warming induced changes in the simulated typhoon intensities. Our methodological proof-of-concept demonstrates the potential of combining the storyline approach with the pseudo-global warming approach, allowing us to frame risk in an event-oriented manner and eventually providing a physical basis for partitioning uncertainties.

We thank the Reviewer for positively providing us with the opportunity to enhance the quality of the manuscript.

**Comment:** Of note, the data availability requirements of HESS (https://www.hydrology-and-earth-system-sciences.net/policies/data_policy.html) do not appear to be satisfied by the statement "All data used to conduct this study will be made available upon request." The simulation data (or at least a subset of it able to recreate the results) should be uploaded to Zenodo or some other repository. Given the relatively small nature of the simulations (i.e., regional domain and short time windows), this shouldn't be an issue with some careful consideration of exactly what data should be archived.
**Reply:** We thank the Reviewer for introducing us to the Data Policy of HESS and the channels to archive the outputs of our study.

In this revision we adhere closely to the Data policy of HESS. Specifically, we upload our model setup (i.e., namelist files), visualization scripts, and key model outputs to the Zenodo repository to support open science.

**Major comments:**
**Comment:** There isn't an objective metric for choosing 7 of the 28 storms. The authors describe evaluations of intensity and track, but only superficially and rapidly jump from Fig. 4 (all TCs) to Fig. 5 (the 7 TC subset). The obvious concern here is -- if the other 21 TCs were poorly enough simulated to not be included in this analysis, is the version of WRF applied here really "fit for purpose"? That is, if it struggles to simulate the TCs, why do we believe the PGW signal is credible? Simply put, I think this decision needs a clearer rationale. If the other storms were excluded due to poor simulation quality, it would be helpful to provide a detailed analysis of these issues and discuss how they might affect the credibility of the remaining PGW simulations.
**Reply:** We utilize objective metrics, i.e., mean bias and typhoon category, to select seven out of the 28 storms for our study. The selection of these seven typhoons for the PGW investigation is based on the evaluation results of model performance in reproducing the observed maximum wind speed and minimum sea level pressure. Given that our study focuses on warming-induced changes in the simulated typhoon intensity, the selected typhoons represent the full range from Category 1 to Category 4. Of course, we are fully aware of the uncertainties in the simulations stemming from model physical schemes, configurations, synoptic situations, and warming scenarios. However, our preliminary simulations indicated that it is challenging to find a single model setup that performs well

across all 28 typhoons. Therefore, our PGW analyses are restricted to the seven selected typhoons, for which our WRF model setup demonstrates the best performance.

We believe that our derived PGW signals are credible, as they are based on the state-of-the-art Coupled Model Intercomparison Project CMIP6. To account for the uncertainties stemming from warming scenarios, we have conducted 18 warming scenario simulations for each typhoon, enabling us to draw our conclusion statistically (Figure 6).

In this revision additional PGW simulations for one selected typhoon (i.e., one with poorer performance) are conducted to further evaluate whether the performance of the WRF model affects the credibility of the remaining PGW simulations.

The above discussions about the selection of analyzed typhoons have been added into the revised "Results" section.

**Comment:** There is no statistical significance testing. This appears to be a fairly big oversight in my mind -- even basic tests like KS or a t-test could shed some light on how robust these changes are. The authors speculate about this occasionally in the manuscript (talking about the "majority" of members that increase in intensity, for example) but more quantitative analysis is needed.
**Reply:** We agree with the Reviewer about the need for statistical significance testing of our results.

In this revision the non-parametric Mann-Whitney U-test is used for quantitative significance testing, since the Gaussian distribution could not be assumed for all results in our study.

**Comment:** The IKE section is very underdeveloped (only 15 lines in the text). The evaluation is very superficial and, in my opinion, could be improved with fairly little effort. For example, Neoguri has a larger signal in the maximum wind versus the IKE. This would imply the structure of the TC is compensating (smaller?) for the IKE to change little even in the face of a large change in wind speed. Conversely, the IKE signal seems larger for Usagi, implying that the storm wind field is expanding (either in the inner core or the outer reaches). This comes back to my overarching concern that this paper reads very much as a shallowish description of model simulations without some deeper probing.
**Reply:** We thank the Reviewer for the guidance in enhancing the quality of our manuscript.

In this revision the IKE section is expanded in more detail. Following the Reviewer's suggestion, warming-induced changes in the structure of typhoons are investigated, such as Neoguri and Usagi, using metrics like vertical velocity and relative vorticity. Based on the findings from this comparative analysis, a similar approach is adopted to deepen our understanding of other typhoon PGW simulations derived from individual CMIP6 models.

**Comment:** In general, the paper could be better served by providing at least some spatial evaluation (e.g., 10.1038/s41586-018-0673-2). For example, is the precipitation maximum just increasing because all precipitation rates are increasing, or is the fundamental structure of the TC changing? In the storms that weaken in the PGW runs, does this appear to just be internal variability in storm intensity (see literature surrounding rapid intensification and weakening in TCs due to inner core processes) or is this fundamentally a response to the large-scale environment (perhaps an evaluation of metrics such as maximum potential intensity would be worthwhile)?
**Reply:** Thank you for sharing the references.

In the revised version, two new analyses are performed:
- A new analysis of changes in the spatial patterns of typhoon-induced precipitation under warming conditions is performed. This analysis helps us determine whether the increase in maximum precipitation is a systematic trend or is related to changes in the structure of the typhoons.
- A new analysis comparing intensified typhoon PGW runs with weakening typhoon PGW runs is performed using metrics, for example, maximum potential intensity. This comparison

provides insights into the physical mechanisms behind the differing responses (i.e., intensification versus weakening) of typhoons under warming conditions.

**Minor comments:**
**Comment:** Lines 106-107. Are we sure CMIP6 versus CMIP5 provides value-added for PGW runs?
**Reply:** We have rephrased this sentence as follows:
"In particular, our study emphasizes the use of the latest CMIP6 ensembles and the strategy of aligning these ensembles for future projections of tropical cyclones."

**Comment:** Fig. 2. While there is a sample of 16 GCMs. There really should be a discussion of model independence -- for example, there are multiple versions of CMCC, MPI, and NorESM2, with the only apparent difference being resolution. I would expect these sets of models to have very similar changes when compared to models with vastly different structural characteristics.
**Reply:** Yes, the inclusion of multiple versions of CMCC, MPI, and NorESM2 in our study facilitates comparisons of the simulations both within similar GCM frameworks and across different GCM frameworks.

In the revision we discuss our results in terms of model dependence and independence to address the uncertainties stemming from the GCMs. For the model dependence analysis, the simulations of CMCC, INM, MPI, and NorESM2 are selected. For the model independence analysis, the simulations of CanESM5 and CAMS-CSM1-0 are selected.

**Comment:** Line 229: 5 km grid spacing with cumulus convection remaining on? At these grid spacings, do researchers typically apply cumulus parameterizations or turn them off?
**Reply:** Yes, we turned on the Kain-Fritsch (KF) cumulus scheme for the 5-km grid spacing simulations.

We acknowledge that a grid spacing of 5 km is generally considered to be within the gray zone for simulating precipitation. This implies that the necessity for a cumulus scheme at this resolution can vary depending on the specific case. For example, Arnault et al. (2019) turned off cumulus parameterization in WRF to accurately reproduce a high-precipitation event in the upper Danube River basin in Europe. In contrast, Delfino et al. (2022) enabled the KF scheme while simulating Typhoon Haiyan over the North West Pacific using WRF.

In our study, we have conducted a series of preliminary simulations with the KF scheme both enabled and disabled. Our results indicate that the WRF model performs better when the KF scheme is enabled, which is in line with findings from sensitivity studies on the WRF model (Delfino et al., 2022; Sun et al., 2024).

The above discussion about the necessity of turning the KF scheme on at a 5 km grid spacing for simulating typhoons has been added into the revised "2.3 Methods" section.

**Reference:**
Arnault, J., Wei, J., Rummler, T., Fersch, B., Zhang, Z., Jung, G., et al., 2019: A joint soil-vegetation-atmospheric water tagging procedure with WRF-Hydro: Implementation and application to the case of precipitation partitioning in the upper Danube River basin. Water Resources Research, 55, 6217–6243. https://doi.org/10.1029/2019WR024780

Sun, Q., Olschewski, P., Wei, J., Tian, Z., Sun, L., Kunstmann, H., and Laux, P., 2024: Key ingredients in regional climate modelling for improving the representation of typhoon tracks and intensities, Hydrol. Earth Syst. Sci., 28, 761–780, https://doi.org/10.5194/hess-28-761-2024

Delfino, R. J., Bagtasa, G., Hodges, K., and Vidale, P. L., 2022: Sensitivity of simulating Typhoon Haiyan (2013) using WRF the role of cumulus convection, surface flux parameterizations, spectral

nudging, and initial and boundary conditions, Nat. Hazards Earth Syst. Sci., 22, 3285–3307, https://doi.org/10.5194/nhess-22-3285-2022

**Comment:** Fig. 4. The authors should include a discussion of how the pressure-wind relationship simulated in models impacts these results (e.g., doi:10.6057/2018TCRR04.01)
**Reply:** Yes, the pressure-wind relationship in typhoons establishes a statistically meaningful link between the environmental conditions and the center of the typhoon, as well as the increase in the maximum surface wind around the storm (Bao et al., 2012).

In this revision the pressure-wind relationship in our typhoon simulations is investigated, and the derived results are added into the "Results" section.

**Reference:**
Bao, J.W., Gopalakrishnan, S.G., Michelson, S.A., Marks, F.D., Montgomery, M.T., 2012, Impact of physics representation in the HWRFX on simulated hurricane structure and pressure-wind relationships, Monthly Weather Review, 140, 3278-3299, https://doi.org/10.1175/MWR-D-11-00332.1

**Comment:** Fig. 6. I assume for the precipitation rates (c,d), the authors first find a single value at each model timestep (e.g., an array that is 1 x ntimes for each member) and then include those in the statistics. That is, there is no inclusion of spatial components in these boxplots?
**Reply:** We do incorporate spatial information when deriving the statistics shown in Figure 6. For instance, the maximum 1-hour precipitation rates (Figure 6c) represent the local maximum values surrounding the storm center within the 4° × 4° evaluation region.

The explanation of our analysis strategy is added in the "Data and Methods" section.

**Comment:** Figs. 7-8. It is very difficult to interpret these results outside of the thicker black, red, and blue lines. If the authors would like to provide more model-specific context, I would suggest either converting these lines to some form of shading or larger figures that could provide the ability to see some of the model lines (which currently are stacked very much on top of one another).
**Reply:** We appreciate the Reviewer's feedback.

We have improved the readability of Figures 7-8 as follows.
- Replace the individual lines with shaded bands to better visualize the uncertainties stemming from GCMs;
- Retain selected model lines within the shaded bands for the model independence and dependence analysis, such as the simulations from CanESM5 and CAMS-CSM1-0.

**Comment:** Lines 422. I am not sure what is particularly unique about Hato and Hagupi. Is it that the median value of the PGW runs falls in the 25-75% range of the ERA5?
**Reply:** Here, we are referring to the comparison between the colored circles (representing the warming scenario runs) to the black circle (representing the baseline run) in Figure 6c. The circles represent the maximum of the 1-hour precipitation extremes throughout the analyzed period. Figure 6c shows no clustering effects for Hato and Hagupit, as the colored circles (warming scenario runs) are widely and evenly distributed around the black circle (baseline run). This is what we meant by "no clear picture of a change in precipitation intensity".

Our detailed explanation about Figure 6 has been added into the revised Results for better understanding.

---

## Author Comment (AC4)

**Response to Review 4 (CC1-Jimmy Fung):**

The authors presented a thorough storyline-based analysis over the Pearl River Delta under future conditions using Pseudo-Global Warming (PGW) modeling techniques. They perturbed the historical initial and boundary conditions obtained from ERA5 reanalysis and handed to the regional model WRF according to the climate change signal projected by the CMIP6 ensemble under SSP5-8.5, in which each of the 16 selected CMIP6 models acted as the basis for an individual storyline of future typhoon intensity. Results indicate a general increase in typhoon intensity across all metrics for six of the seven inspected typhoons. This increase is notably higher for specific storylines, and the projected increase in the extreme values of the inspected metrics significantly exceed the median change of all storylines. Overall, the manuscript is well-written and well-organized, and is equipped with a large workload and high-quality numerical experiments as key supporting materials. I believe the method is scientifically valid and the results are logical and reasonable. Based on the current phase of the manuscript, if the authors can properly address the following comments, I would recommend accepting it for publication.

**Reply:** We thank Reviewer 4 for the positive feedback. We are trying our best to address the comments and make the necessary revisions. We hope to have the option to submit a revised version of our manuscript and receive your recommendation for final acceptance.

**Minor Comments:**

**Comment:** The authors utilized CMA and JMA best track data to evaluate the model's performance. Meanwhile, the Hong Kong Observatory (HKO) also provides best track data for tropical cyclones that affect the PRD. Due to Hong Kong's geographical location, HKO's best track data is generally considered to be of high quality for the PRD region. If it does not require a significant amount of extra work, the authors may want to consider comparing their results with HKO's best data in Figures 4 and 5.

**Reply:** We thank Reviewer 4 for sharing with us the information about best track data documented by the Hong Kong Observatory (HKO).

We have downloaded the tropical cyclone best track data by following the link https://data.gov.hk/en-data/dataset/hk-hko-rss-tropical-cyclone-best-track-data.

In this revision our results are compared with HKO's best track data, in addition to CMS's and JMA's best track data, and Figures 4 and 5 are updated accordingly. Discussions about the uncertainties associated with these three best track datasets are added into the "4 Discussion" section.

**Comment:** The CMA best track archives the 2-minute average maximum sustained wind speed at the 10-meter level, while JMA archives 10-minute values. On the other hand, the default output for WRF is the instantaneous wind speed of the grid area mean, which may be related to the integration time step and grid area. It would be helpful if the authors could clarify how they compared these different definitions of wind speed, and whether they used any conversion coefficients.

**Reply:** For our WRF simulations, we saved the default outputs at an hourly scale, which includes, among other variables, instantaneous wind speed and accumulated precipitation. In addition to the default outputs, we saved diagnostics outputs (statistical values) for surface variables such as wind speeds at 2-minute intervals, aligning with the definition of wind speeds documented in the CAM best track. These diagnostics, saved at a high temporal frequency (i.e., 2-minute), allow us to compare the simulated winds speeds with those documented by different agencies in a statistically consistent manner, without needing to apply conversion coefficients.

Our evaluation strategy has been added into the "Data and Methods" section.

**Comment:** The authors employed spectral nudging to enhance the performance of TC tracks. Although the authors have justified that the nudging operation does not affect the inner core of the simulated typhoon, studies have shown that nudging in perturbed experiments may limit the intensity of TCs (Moon et al. 2018; Li et al. 2024). Therefore, it is a trade-off to turn on nudging to reproduce

comparable TC tracks in the PGW experiments. The authors may want to consider discussing this in L250-255.

**Reply:** We fully agree with the Reviewer that using spectral nudging involves a trade-off in reproducing comparable TC tracks in the PGW experiments.

We have rephrased this sentence to better acknowledge the impact of spectral nudging on the simulated typhoons, especially in the PGW experiments. Additionally, we conduct a quantitative analysis of the spatial changes in simulated typhoon tracks due to climate change, focusing particularly on runs without spectral nudging. We also discuss the uncertainties in the results related to the application of spectral nudging.

**Comment:** L165: Please give specific reasons for the preference of using grid-cell basis compared to spatial mean.

**Reply:** The preference for using a grid-cell basis rather than a spatial mean stems from our aim of capturing the spatial (horizontal and/or vertical) variability of climate change signals.

This consideration is important because our simulation domain is relatively large, approximately $2.5*10^7$ km$^2$. Global warming signals vary regionally across such a large domain and differ between global general circulation models as well.

Justifications for our preference for using a grid-cell basis over a spatial mean have been added in the revised "Data and Methods" section.

**Comment:** L175-180: Although seven selected typhoon events did not occur in May and June, it is still valuable to fill the 2-month gap in Figure 2 to give a complete picture of summer-time seasonal cycle of global warming signal for readers' reference.

**Reply:** As suggested, the May-June 2-month gap has been filled in the updated Figure 2 for reader's reference, so that a complete picture of summer-time seasonal cycle of global warming signal is given in the revised version.

**Comment:** L210: The authors have provided a justification for using WRF instead of HWRF. However, it is important to note that the ocean coupling effect, such as cold wake feedback and wind-sea wave interactions, may also impact the intensity of TCs, which cannot be captured by WRF-only simulations (e.g., Mogensen et al. 2017; Magnusson et al. 2019; Li et al. 2022). Therefore, the authors may want to address some of these limitations in their study.

**Reply:** In this revision we have followed the Reviewer's suggestion and have acknowledged the limitations of our WRF-only simulations by expanding on the following aspects:

- Representation of air-wave-ocean interactions;
- The need for coupling with an ocean model and/or wave model;
- Limitations regarding operational real-time hurricane forecasting.

The above-mentioned aspects concerning the limitations of our WRF-only simulations have been further elaborated in the revised "4.3 Limitations of the study design" section.

**Comment:** L230-235: The authors utilized the Rapid Radiative Transfer Model (RRTM) for longwave radiation and the Dudhia scheme for shortwave radiation in their WRF configuration. To the best of my knowledge, RRTM shortwave scheme is typically used in conjunction with the longwave radiation scheme. Therefore, it would be beneficial if the authors could provide a justification for why they did not use the RRTM shortwave scheme in their study.

**Reply:** We agree with the Reviewer that the RRTM shortwave scheme is typically used in conjunction with the RRTM longwave radiation scheme. However, our choice is based on extensive literature research.

Our justifications for using the Dudhia scheme rather than RRTM for shortwave parameterization are as follows:

- Under clear-sky conditions, several WRF model evaluation studies have shown that the simpler Dudhia shortwave radiation scheme outperforms the more complex RRTM shortwave radiation scheme (Ruiz-Arias et al. 2013; Zempila et al. 2016; Chen et al. 2017).
- In cloudy-sky and precipitating conditions, the combination of the Dudhia shortwave and RRTM longwave schemes is commonly used, for example, in reproducing the East Asian monsoon (Wei et al., 2015), the West African monsoon (Klein et al., 2015), and warm-season precipitation over Europe (Arnault et al., 2018).
- Under extreme weather conditions, this radiation combination has been employed to simulate typhoons over the western North Pacific (Sun et al., 2019; Wu et al., 2024) and the North Atlantic (Perez-Alarcon et al., 2024).

We acknowledge that the combination of the RRTM shortwave and longwave schemes is also used for simulating Typhoons (Li et al., 2022). However, we think that the uncertainties steaming from the selection between RRTM and Dudhia are negligible compared to those associated with the choice of for example cumulus schemes (Tian et al., 2021).

Our justifications for using the Dudhia scheme have been added in the revised version.

**References:**
Ruiz-Arias, J. A., J. Dudhia, F. J. Santos-Alamillos, and D. PozoVazquez, 2013: Surface clear-sky shortwave radiative closure intercomparisons in the Weather Research and Forecasting Model. *J. Geophys. Res. Atmos.*, 118, 9901–9913, https://doi. org/10.1002/jgrd.50778

Zempila, M.-M., T. M. Giannaros, A. Bais, D. Melas, and A. Kazantzidis, 2016: Evaluation of WRF shortwave radiation parameterizations in predicting global horizontal irradiance in Greece. *Renewable Energy*, 86, 831–840, https://doi.org/10. 1016/j.renene.2015.08.057

Chen, W.-D., F. Cui, H. Zhou, H. Ding, and D.-X. Li, 2017: Impacts of different radiation schemes on the prediction of solar radiation and photovoltaic power. *Atmos. Oceanic Sci. Lett.*, 10, 446–451, https://doi.org/10.1080/16742834.2017.1394780

Wei, J., H. R. Knoche, and H. Kunstmann, 2015: Contribution of transpiration and evaporation to precipitation: An ET-Tagging study for the Poyang Lake region in Southeast China, *J. Geophys. Res. Atmos.*, *120*, 6845 – 6864, https://doi.org/10.1002/2014JD022975

Klein, C., Heinzeller, D., Bliefernicht, J. *et al.* 2015: Variability of West African monsoon patterns generated by a WRF multi-physics ensemble. *Clim Dyn* 45, 2733–2755 https://doi.org/10.1007/s00382-015-2505-5

Arnault, J., Rummler, T., Baur, F., Lerch, S., Wagner, S., Fersch, B., Zhang, Z., Kerandi, N., Keil, C., & Kunstmann, H., 2018: Precipitation sensitivity to the uncertainty of terrestrial water flow in WRF-hydro: An ensemble analysis for Central Europe. *Journal of Hydrometeorology*, 19(6), 1007–1025. https://doi.org/10.1175/jhm-d-17-0042.1

Perez-Alarcon, A., Vazquez, M., Trigo, R.M., Nieto, R., Gimeno, L., 2024. Evaluation of WRF model configurations for dynamic downscaling of tropical cyclones activitiy over the North Atlantic basin for Lagrangian moisture tracking analysis in future climate. *Atmos. Res.* 307, 107498. https://doi.org/10.1016/j.atmosres.2024.107498

Sun, J., He, H., Hu, X., Wang, D., Gao, C., Song, J., 2019: Numercial Simulations of Typhoon Hagupit (2008) Using WRF. *Weather and Forecasting*, 34(4), 999–1015. https://doi.org/10.1175/WAF-D-18-0150.1

Wu, J., Gao, L., Meng, Q., Wang, H., 2024. Effect of land cover pattern on rainfall during a landfalling typhoon: A simulation of Typhoon Hato. *Atmos. Res.* 303, 107329. https://doi.org/10.1016/j.atmosres.2024.107329

Li, Z., Tam, C.Y., Li, Y., Lau, N.C., Chen, J., Chan, S.T., Dickson Lau, D.S. and Huang, Y., 2022. How Does Air-Sea Wave Interaction Affect Tropical Cyclone Intensity? An Atmosphere-Wave-Ocean Coupled Model Study Based on Super Typhoon Mangkhut. *Earth and Space Science*, 9(3), https://doi.org/10.1029/2021EA002136

Tian, J., Liu, R., Ding, L., Guo, L., Liu, Q., 2024. Evalution of the WRF physical parameterizations for Typhoon rainstrom simulation in southeast coast of China. *Atmos. Res.* 247, 105130. https://doi.org/10.1016/j.atmosres.2020.105130

**References:**
Li, Z., Tam, C.Y., Li, Y., Lau, N.C., Chen, J., Chan, S.T., Dickson Lau, D.S. and Huang, Y., 2022. How Does Air-Sea Wave Interaction Affect Tropical Cyclone Intensity? An Atmosphere-Wave-Ocean Coupled Model Study Based on Super Typhoon Mangkhut (2018). Earth and Space Science, 9(3), p.e2021EA002136.

Li, Z., Fung, J.C., Wong, M.F., Lin, S., Cai, F., Lai, W. and Lau, A.K., 2024. Future changes in intense tropical cyclone hazards in the Pearl River Delta region: an air-wave-ocean coupled model study. Natural Hazards, pp.1-16.

Magnusson, L., Bidlot, J.R., Bonavita, M., Brown, A.R., Browne, P.A., De Chiara, G., Dahoui, M., Lang, S.T.K., McNally, T., Mogensen, K.S. and Pappenberger, F., 2019. ECMWF activities for improved hurricane forecasts. Bulletin of the American Meteorological Society, 100(3), pp.445-458.

Mogensen, K.S., Magnusson, L. and Bidlot, J.R., 2017. Tropical cyclone sensitivity to ocean coupling in the ECMWF coupled model. Journal of Geophysical Research: Oceans, 122(5), pp.4392-4412.

Moon, J., Cha, D.H., Lee, M. and Kim, J., 2018. Impact of spectral nudging on real- time tropical cyclone forecast. Journal of Geophysical Research: Atmospheres, 123(22), pp.12-647.

**Reply:** The recommended literatures have been cited in the revised version.